# Remember the Past: Distilling Datasets into Addressable Memories for Neural Networks

**Zhiwei Deng**   **Olga Russakovsky**
Department of Computer Science
Princeton University
{zhiweid, olgarus}@cs.princeton.edu

## Abstract

We propose an algorithm that compresses the critical information of a large dataset into compact addressable memories. These memories can then be recalled to quickly re-train a neural network and recover the performance (instead of storing and re-training on the full original dataset). Building upon the dataset distillation framework, we make a key observation that a *shared common representation* allows for more efficient and effective distillation. Concretely, we learn a set of bases (aka "memories") which are shared between classes and combined through learned flexible addressing functions to generate a diverse set of training examples. This leads to several benefits: 1) the size of compressed data does not necessarily grow linearly with the number of classes; 2) an overall higher compression rate with more effective distillation is achieved; and 3) more generalized queries are allowed beyond recalling the original classes. We demonstrate state-of-the-art results on the dataset distillation task across six benchmarks, including up to 16.5% and 9.7% in retained accuracy improvement when distilling CIFAR10 and CIFAR100 respectively. We then leverage our framework to perform continual learning, achieving state-of-the-art results on four benchmarks, with 23.2% accuracy improvement on MANY. The code is released on our project webpage[1].

## 1  Introduction

Compressing a large amount of information into a small memory storage space is one of the key components of human intelligence [1–3] – a person can retrieve memories from the past and quickly recover the corresponding skills. Deep learning methods have made large strides in building task-specific models, but are shown to easily forget past knowledge when learning new tasks [4, 5].

To equip neural network learners with memorizing ability, dataset distillation [6] is proposed as a potential solution. Concretely, a compressed set of examples (memories) is learned to summarize the key information in a dataset that affects model training; these examples can then be used to quickly retrain models and recover the corresponding skills. This differs from the standard reconstruction-based compression algorithms [7–9] and shows strong performance [10–13].

A critical question in building powerful compressed memories is: what structures and representations should we use to build the memories? An effective structure and organization of memories can lead to different fundamental assumptions about data and affect the compression and learning behaviours. Existing works [10, 11, 14, 12, 15, 13, 6] follow a simple representation, where a set of learnable examples is assigned for each class. However, under this assumption, the size of the memories can linearly grow with the number of classes, making the distillation of datasets with a large number of classes challenging. Naturally, this can potentially lead to redundancies in the learned memories,

---

[1] https://github.com/princetonvisualai/RememberThePast-DatasetDistillation

36th Conference on Neural Information Processing Systems (NeurIPS 2022).

due to the separation of data among classes. Furthermore, this representation is less generalizable to continuous label space, where infinite number of label values exists.

In our paper, we make the observation that there is information shared between classes, and hypothesize that a common and compact representation exists for all classes. Following this hypothesis, we propose to formulate the problem as a memory addressing process, where the memories store a common set of bases shared by all classes, and the recombination of bases is performed through an addressing function. This decomposition between memories and addressing functions enables the possibility that all common information is stored in one part of the representation, and the accessing of the common information depends on the specific labels and is handled through an extra function. We find that this formulation can significantly improve both the compression rate and the performance.

We adopt the back-propagation through time learning framework to train the memories and addressing functions, and identify several critical factors that can improve the performance. Specifically, we find that adopting the momentum term, and performing long unrolls in the inner optimization loop are both critical. This differs from the common usage of bi-level optimization algorithm on this task [6, 16], and leads to strong performance outperforming single-step gradient matching methods [10, 11] even with the simple data representation.

In the experiments, we extensively evaluate our algorithm on six benchmarks of the Dataset Distillation task, and show that it consistently outperforms previous state-of-the-art by a significant margin. For example, we achieve 66.4% accuracy on CIFAR10 with the storage space of 1 image per class, improving over the previous state-of-the-art KIP method [12, 13] by 16.5%. We further demonstrate our method on the continual learning tasks, and show that a simple "compress-then-recall" method using our framework leads to state-of-the-art results on four datasets. For example, we outperform all prior methods by 23.2% in retained accuracy on the challenging MANY [17] benchmark. Finally, we demonstrate the generality of our approach by extending to image-based (rather than label-based) memory recall, and synthesizing new classifiers (unseen during training) from our distilled memories.

## 2 Related works

**Dataset Distillation.** The task of dataset distillation is fundamentally a compression problem, with a different prioritization on the information contained in data. There have been several lines of methods, developed with different criteria to prioritize information. *Generalization loss* with bi-level optimization framework [18–20] has been widely studied and is used in the early works of dataset distillation [6, 16]. It emphasizes on the loss at the final optimization state. *Gradient-matching or score-matching* methods [10, 11, 15] are adopted to directly match the induced gradients from synthetic data. If ideally matched over the gradient field, the compressed dataset can naturally lead to the same model parameters with gradient descent. *Kernel method* [13, 12] shows that with the connection to Gaussian processes, a kernel inducing points method can be used to achieve strong performance, but with large computation costs. These are also connected with the recent progress on pragmatic compression methods [21, 22], which compress or match distributions based on a decision process (in dataset distillation's case, the gradient descent search process).

**Continual learning.** Broadening the learning paradigms, continual learning problem aims to build agents that learn through a stream of tasks and accrue knowledge along the process. "Catastrophic forgetting" [5, 23, 4] is a well-known phenomenon in this setting, where the neural network forgets previous skills when learning new ones. Various methods [24–27, 17, 28–35] on regularization, replay, or dynamic model, have been proposed to alleviate the issue and address the "stability-plasticity dilemma" [36, 37]. Memory buffer has been a critical component in the past methods [24–27, 17, 28, 38, 39], but mainly relies on a random selection of real samples with different strategies. Recently, several works extend the usage of memory to storing random basis [39] (online setting) or SVD bases [38] (offline setting).

## 3 Background: dataset distillation

The task of Dataset Distillation [6] is proposed to compress the key information of a large-scale training dataset into a small amount of learned data, which can be stored using limited memory space and retrieved through label indices or task information to recover the performance of a model.

**Problem Setting.** Formally, given a large dataset $\mathcal{D}_{tr} = \{(\boldsymbol{x}_i, \boldsymbol{y}_i)\}_{i=1}^{N}$ containing $N$ pairs of training data $(\boldsymbol{x}_i, \boldsymbol{y}_i)$, where $\boldsymbol{x}_i$ is an image and $\boldsymbol{y}_i$ is the corresponding label in $C$ classes, a small dataset $\mathcal{D}_s = \{(\boldsymbol{x}'_j, \boldsymbol{y}_j)\}_{j=1}^{N'}, N' \ll N$, can be synthesized or distilled, such that a model trained on $\mathcal{D}_s$ can have the same generalization ability as ones trained on $\mathcal{D}_{tr}$:

$$\mathbb{E}_{(\boldsymbol{x},\boldsymbol{y})\sim\mathcal{D}_{te}}\left[\mathbf{m}(f(\boldsymbol{x};\boldsymbol{\theta}^{(*)}),\boldsymbol{y})\right] \simeq \mathbb{E}_{(\boldsymbol{x},\boldsymbol{y})\sim\mathcal{D}_{te}}\left[\mathbf{m}(f(\boldsymbol{x};\boldsymbol{\theta}'^{(*)}),\boldsymbol{y})\right] \tag{1}$$

where $\boldsymbol{\theta}^{(*)}$ and $\boldsymbol{\theta}'^{(*)}$ are the optimized parameters using $\mathcal{D}_{tr}$ and $\mathcal{D}_s$ respectively, $\mathbf{m}$ is a metric, e.g., accuracy, and $\mathcal{D}_{te}$ is the test dataset. The model is often a neural network classifier $f(\cdot;\boldsymbol{\theta})$ parameterized by $\boldsymbol{\theta}$ and trained with a loss function $\ell(f(\boldsymbol{x};\boldsymbol{\theta}),\boldsymbol{y})$.

**Synthetic dataset representations.** The synthetic dataset contains the core information that needs to be learned. The representation of the synthetic data affects the compactness and effectiveness of the distillation process. In existing methods [6, 10, 11, 14, 12, 15], the dataset $\mathcal{D}_s$ is defined as a collection of learnable data samples $(\boldsymbol{x}', \boldsymbol{y})$, and the number of samples is separately and equally distributed across classes. This representation has several disadvantages: first, the number of synthetic data samples needed for a dataset grows linearly with the number of classes, leading to limited applicability when the number of classes is large or undefined (e.g., language or other continuous labels); second, the potentially shared and common information across classes is ignored – this results in a less compact representation of the distilled information and lower compression rate; lastly, the representation is not able to generalize to new classes or tasks, due to the lack of common representation learned across classes.

# 4 Model

**Overview.** In this section, first, we present a new perspective of the problem, where the Dataset Distillation problem is formulated as a *memory addressing process*: instead of learning synthetic images separately for each class, we construct and learn a common memory representation that can be accessed through addressing matrices to construct synthetic datasets. Under this formulation, the number of synthetic images does not need to grow linearly with the number of classes, the shared information among classes can be exploited to reduce redundancies and improve compression rate, and datasets can be distilled with respect to more generic queries. Second, we further show several critical empirical facets of back-propagation through time framework, which lead to drastic improvements on the performance and outperform the single-step gradient matching methods. This is in contrast with the current common observation that gradient matching outperforms back-propagation through time framework on dataset distillation tasks.

In the following, we present the two core components of our method, (1) the new formulation of dataset distillation, with memories and addressing matrices in Sec. 4.1, and (2) the learning framework under back-propagation through time in Sec. 4.2.

## 4.1 Dataset Distillation as memory addressing

**Problem formulation.** Given a task-specific dataset $\mathcal{D}_{tr} = \{\boldsymbol{x}_i, \boldsymbol{y}_i\}_{i=1}^{N}, \boldsymbol{x} \in \mathcal{X}, \boldsymbol{y} \in \mathcal{Y}$, we aim to learn a single compact and compressed representation $\mathcal{M}$, referred to as memories, that can be accessed through a learned addressing function $\mathcal{A}(\cdot)$. $\mathcal{A}(\cdot)$ takes all possible values of $\boldsymbol{y}$ as input and recalls the corresponding synthetic data. With a set of $\{\boldsymbol{y}_i\}$, a synthetic dataset recalled using the above process can train a model $f_{\boldsymbol{\theta}} : \mathcal{X} \to \mathcal{Y}$ from scratch and obtain the same generalization ability as trained on $\mathcal{D}_{tr}$.

Ideally, the memories $\mathcal{M}$ and the addressing function $\mathcal{A}$ can jointly capture the critical information that defines the task mapping from $\mathcal{X}$ to $\mathcal{Y}$, such that the recalled synthetic data given a query $\boldsymbol{y}_i$ contains distinctive information that defines $\boldsymbol{y}_i$ and the synthetic dataset recalled with $\{\boldsymbol{y}_i\}$ can satisfy eqn. 1 when used for re-training. For example, in the standard classification tasks, we can enumerate all possible values in the label space (discrete) and address the memories, to construct the synthetic dataset that contains critical information. Under this formulation, since we are learning a single, shared and accessible representation for all $\boldsymbol{y}$s, the size of memories can be defined flexibly regardless of the number of classes, removing the linear growth limitation in the standard distillation settings. There is also no constraint on the form of $\boldsymbol{y}$s, which can be either discrete or continuous. The *storage budget* or *compression rate* is calculated by considering the storage space of parameters in both memories and addressing functions, which should be as compact as possible.

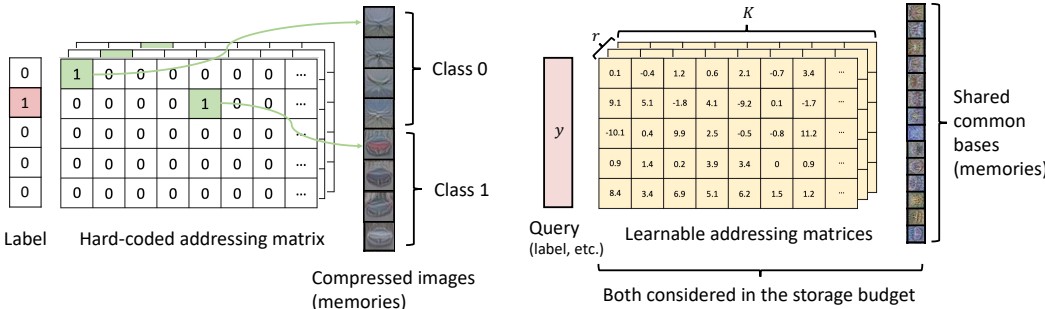

Figure 1: Distilling a large-scale dataset into compressed memories. *Left:* the standard dataset distillation task under the formulation of memory addressing. The addressing matrices are hard-coded with 1s and 0s to fetch the corresponding compressed image in memories. The memory size grows linearly with number of classes. *Right:* learnable addressing matrix with shared common bases (number of bases can be flexibly defined). The information sharing between classes is captured in this representation. The queries can be generalized to any vector representation, besides one-hot labels, i.e. for a general dataset from $\mathcal{X}$ to $\mathcal{Y}$, it can be distilled into memories for recall and model re-training.

**Memory representation.** We use a set of bases to store in the memories $\mathcal{M} = \{\boldsymbol{b}_1, ..., \boldsymbol{b}_K\}$, where each vector $\boldsymbol{b}_k \in \mathbb{R}^d$ has the same dimension as $\boldsymbol{x} \in \mathbb{R}^d$, and all vectors collectively define the intrinsic components in a dataset that characterizes the task mapping from $\mathcal{X}$ to $\mathcal{Y}$. Through re-using the bases, we can produce a desired synthetic dataset for model re-training. *Spatial redundancies in images:* note that, as a special case, images can contain redundancies spatially and be stored in a downsampled version to improve the compression rate. The downsampled image bases can be passed via a deterministic upsampling process (e.g., bilinear interpolation) to recover the original resolution.

**Memory addressing.** For each query $\boldsymbol{y}$, we use a parameterized function $\mathcal{A}(\boldsymbol{y})$ to re-combine the bases in the memories $\mathcal{M}$. Similar to previous methods on accessing memories, we use $\mathcal{A}$ to produce a set of coefficients, and linearly combine the bases to produce synthetic data. Formally, to retrieve $r$ synthetic examples for each $\boldsymbol{y}$, we define a set of matrices $\{\boldsymbol{A}_1, ..., \boldsymbol{A}_r\}$, $\boldsymbol{A}_i \in \mathbb{R}^{d_y \times K}$, where $d_y$ is the dimension size of $\boldsymbol{y}$ and $r$ is the number of data samples that can be retrieved. With the memories $\mathcal{M} = \{\boldsymbol{b}_1, ..., \boldsymbol{b}_K\}$, we define:

$$\boldsymbol{x}_i'^T = \boldsymbol{y}^T \boldsymbol{A}_i [\boldsymbol{b}_1; ...; \boldsymbol{b}_K]^T, \boldsymbol{x}' \in \mathbb{R}^{d \times 1} \tag{2}$$

where $\boldsymbol{y} \in \mathbb{R}^{d_y \times 1}$ is in a vectorized form, such one-hot encoding of categorical labels, and $\boldsymbol{v} = \boldsymbol{y}^T \boldsymbol{A}_i$ corresponds to a coefficient vector $\boldsymbol{v}$ that combines the bases. The produced synthetic data $\boldsymbol{x}'$ is paired with $\boldsymbol{y}$ as the corresponding label. Our model is shown on the right of figure 1.

**Constructing a dataset.** To construct a synthetic dataset $\mathcal{D}_s$ for model re-training, we are often given a set of samples $\{\boldsymbol{y}_i\}$, or can enumerate all possible values of $\boldsymbol{y}$ (if discrete). With the set $\{\boldsymbol{y}_i\}$, we use eqn.2 to address and retrieve the synthetic dataset $\mathcal{D}_s^{\boldsymbol{y}=\boldsymbol{y}_i} = \{(\boldsymbol{x}_j', \boldsymbol{y}_i)\}_{j=1}^r$ for each $\boldsymbol{y}_i$. The final dataset is the union of all $\mathcal{D}_s = \bigcup \mathcal{D}_s^{\boldsymbol{y}=\boldsymbol{y}_i}$. The dataset $\mathcal{D}_s$ can be used in either a minibatch form for stochastic gradient descent, or as a whole for batch gradient descent.

**Generalized possibilities of queries.** Another advantage of our formulation, besides a compact and shared representation, is the various possibilities of queries $\boldsymbol{y}$. In principle, under this formulation, the label $\boldsymbol{y}$ can be flexibly defined as other forms, such as language or audio, where the representation resides in a continuous space or follows a distribution $p(h(\boldsymbol{y}))$ defined by a feature extractor $h(\cdot)$. The set of $\{\boldsymbol{y}_i\}$ then can be sampled from the distribution $p$ instead of having to enumerate all possible values. This provides a general way of compressing or distilling a large dataset without constraint on the forms of labels.

**Connection with standard Dataset Distillation.** In the standard setting, dataset distillation is defined for classification tasks with discrete labels and each label owns its unique set of synthetic data. We can show that this is a special case of our formulation: if the bases are constructed as the collection of those label-specific synthetic data ($K = N'$, $N'$ is the total size of the synthetic dataset), the addressing matrices $\boldsymbol{A}_i$ are defined in space $\{0, 1\}^{C \times N'}$ where $\boldsymbol{A}_i[m, n] = 1$ if $n$ equals

**Algorithm 1**

1: **hyperparameters:** Momentum rate $\beta_0, \beta_1$, learning rate $\alpha_0, \alpha_1$ for $\boldsymbol{\theta}$ and $\boldsymbol{\phi}$ respectively.
2: **input:** Dataset $\mathcal{D}_{tr}$, memories $\mathcal{M}$, addressing function $\mathcal{A}$, loss function $\ell(\cdot, \cdot)$
3: **repeat**
4:     Sample a subset of labels $\mathcal{Y}'$
5:     Address the memories $\mathcal{M}$ and obtain synthetic dataset $\mathcal{D}_s^{\mathcal{Y}'}$ with eqn. 2
6:     Randomly initialize model parameters $\boldsymbol{\theta}_0$
7:     Initialize momentum $\boldsymbol{m}_0 = 0$
8:     **for** $t = 1$ **to** $T$ **do**
9:         Sample a minibatch $\boldsymbol{B}_s = \{(\boldsymbol{x}_i', \boldsymbol{y}_i)\}$ from $\mathcal{D}_s^{\mathcal{Y}'}$
10:         Compute $\mathcal{L} = \frac{1}{|\boldsymbol{B}_s|} \sum_{i=1}^{|\boldsymbol{B}_s|} \ell(f_{\boldsymbol{\theta}_{t-1}}(\boldsymbol{x}_i'), y_i)$
11:         **Update momentum** $\boldsymbol{m}_t = \beta_0 \boldsymbol{m}_{t-1} + \frac{d\mathcal{L}}{d\boldsymbol{\theta}_{t-1}}$
12:         Update $\boldsymbol{\theta}_t = \boldsymbol{\theta}_{t-1} - \alpha_0 \boldsymbol{m}_t$
13:     **end for**
14:     Sample a minibatch $B = \{(\boldsymbol{x}_i, \boldsymbol{y}_i)\}$ from $\mathcal{D}_{tr}$ with labels in $\mathcal{Y}'$
15:     Compute $J(\boldsymbol{\phi}) = \frac{1}{|\boldsymbol{B}|} \sum_{i=1}^{|\boldsymbol{B}|} \ell(f_{\boldsymbol{\theta}_T}(\boldsymbol{x}_i), \boldsymbol{y}_i)$
16:     Update $\boldsymbol{\phi} \leftarrow \text{OPT-STEP}(\boldsymbol{\phi}, J(\boldsymbol{\phi}), \alpha_1, \beta_1)$
17: **until** Converge

---

to $m(N'/C) + i$ ($i^{th}$ item of $m^{th}$ class) and $\boldsymbol{A}_i[m, n] = 0$ at other positions, then the "retrieval" process can also be defined as eqn. 2.

## 4.2 Learning framework: back-propagation through time

In this section, we build upon the back-propagation through time algorithm and discuss in detail the learning framework that performs the distillation process from a dataset to memories and addressing functions.

Starting from notations, we define the parameters contained in both memories and addressing functions as $\boldsymbol{\phi}$, which are collectively optimized. A loss function is $\ell(\cdot, \cdot)$ is defined on a task-specific dataset. We denote an optimization algorithm as $\text{OPT}(\cdot, \cdot; \alpha, \beta, \ell, T)$, where $\alpha$ and $\beta$ are the learning rate and the momentum rate, respectively, and $T$ is the number of optimization steps. For a single step optimization, we denote it as $\text{OPT-STEP}(\cdot, \cdot; \alpha, \beta)$.

To learn the parameters $\boldsymbol{\phi} = \{\mathcal{M}, \mathcal{A}\}$, we follow a standard bi-level optimization framework with back-propagation through time (BPTT), where the inner-loop uses the synthetic dataset $\mathcal{D}_s$ to train a randomly initialized model starting from scratch, and a generalization loss is computed using a minibatch $\boldsymbol{B} = \{(\boldsymbol{x}_i, \boldsymbol{y}_i)\}$ sampled from $\mathcal{D}_{tr}$. The parameters $\boldsymbol{\phi}$ are implicitly contained in the synthetic dataset $\mathcal{D}_s$ and optimized when minimizing the generalization loss. The bi-level optimization defines:

$$\min J(\boldsymbol{\phi}) = \frac{1}{|\boldsymbol{B}|} \sum_{i=1}^{|\boldsymbol{B}|} \ell(f(\boldsymbol{x}_i; \boldsymbol{\theta}^*), \boldsymbol{y}_i),$$
$$\text{subject to} \quad \boldsymbol{\theta}^* = \text{OPT}(\boldsymbol{\theta}_0, \mathcal{D}_s; \alpha_0, \beta_0, \ell, T) \tag{3}$$

where $\boldsymbol{\theta}_0$ represents the initializing parameters, $\boldsymbol{\theta}^*$ is the optimized model parameters in the inner optimization loop, $\alpha_0$ and $\beta_0$ are the learning rate and momentum rate for $\text{opt}(\cdot, \cdot)$, and $J(\boldsymbol{\phi})$ is the generalization loss on minibatch $\boldsymbol{B}$. In practice, for each inner loop training, we randomly sample a subset $\mathcal{Y}'$ from $\boldsymbol{y}$s and retrieve the corresponding subset $\mathcal{D}_s^{\mathcal{Y}'}$. This reduces the computation cost in inner loops. We empirically observe that equivalent results can be achieved with faster runtime. The algorithm is summarized in Alg. 1, where lines 8-13 define the inner loop optimization process $\text{OPT}(\cdot, \cdot; \alpha, \beta, \ell, T)$, and the gradients of generalization loss (line 15) is back-propagated through the inner loop to update $\boldsymbol{\phi}$. Note that, in principle, the inner loop optimization can be performed using any optimizer. In this paper, we mainly rely on the standard stochastic gradient descent with momentum to train the distilled data.

**Critical factors in BPTT.** Although being a natural choice in performing dataset distillation and adopted in the original work [6], the BPTT framework has been shown to underperform other algorithms, such as single-step gradient matching methods [10, 11], on various benchmarks. The underlying causes that hinder the performance of the algorithm are still underexplored. In our work, we investigate and identify the factors that can unleash the potential of back-propagation through time framework on dataset distillation and lead to significant performance boosts.

*Momentum term.* In previous dataset distillation works [6, 16], the usage of back-propagation through time framework omits the momentum term in the inner loop optimization. Indeed, this has been a common practice in meta-learning tasks [18, 40]. Adding momentum terms in meta-learning can potentially even hurt the performance and lead to less gradient diversity [40]. However, we observe that, in dataset distillation tasks, the momentum term is crucial for making BPTT excel, even in the relatively short inner loop optimization settings (e.g. 10 or 20 steps). We provide results and analysis in the experiments section.

*Long unrolled trajectories.* Another aspect in using BPTT in dataset distillation is the length of unrolled optimization trajectories in the inner loops. The previous usage of BPTT on this task [6, 16] adopts relatively short inner loop optimization trajectories (e.g. 10-30 steps). Instead, we show that unrolling the trajectories long enough (e.g. 200 steps) with momentum terms can potentially produce $\theta^*$ that better summarizes the information contained in memories and addressing matrices, generating more effective gradients to learn the compressed representation.

## 5 Experiments

We thoroughly evaluate our model and demonstrate the benefits over previous methods. In section 5.1, we show that using a shared representation is critical to improving the distillation performance and compression rates. Specifically, we observe that there is strong evidence that there is information re-using across classes. We further show the benefits of our model on standard continual learning tasks in section 5.2. For example, we observe that a simple "compress-then-recall" method can achieve performance outperforming state-of-the-art continual learning models with complex designs. Finally, in section 5.3, we show that storing the compressed data enables synthesizing new classifiers (section 5.3.1) and the shared representation formulation allows more general queries (section 5.3.2), which can be continuous (e.g. image features).

### 5.1 Dataset Distillation

In this section, we follow the standard setting of dataset distillation, and perform dataset compression that can be recalled with discrete class labels.

**Datasets.** We test our models on six standard dataset distillation benchmarks: MNIST [41], FashionMNIST [42], SVHN [43], CIFAR10 [44], CIFAR100 [44], and TinyImageNet [45]. MNIST contains 10 classes with 60,000 writing digit images as training and 10,000 as test set. The images are gray-scale with a shape of $28 \times 28$. FashionMNIST is a dataset with clothing and shoe images and consists of a training with size 60,000 and a test set with size 10,000. Each image is $28 \times 28$ in gray scale, and has a label from 10 classes. SVHN contains street digit images where each image has shape $32 \times 32 \times 3$. The dataset contains 73,257 images for training and 26,032 images for testing. CIFAR10 and CIFAR100 are color image datasets, with 50,000 training images and 10,000 testing images on each. CIFAR10 has 10 classes with 5,000 images per class, and CIFAR100 has 100 classes with 500 images per class. TinyImageNet [45] contains 200 categories with images of resolution 64x64. The training and testing sets have 100,000 and 10,000 images respectively.

**Experiment settings.** We evaluate our distillation models under three different memory budgets for each dataset: 1/10/50 images per class. We focus on high compression rate scenarios and consider the 1 and 10 settings for CIFAR100. Following previous works [14, 11, 12, 15], the main network architecture used in experiments is a simple convolutional network (ConvNet) with $3 \times 3$ filters, InstanceNorm, ReLU and $2 \times 2$ average pooling. For the evaluation protocol, each model is evaluated on 20 randomly initialized models, trained for 300 epochs on a synthetic dataset, and tested on a held-out testing dataset. We use one GPU per experiment run.

**Memory budget calculation.** Since our model uses memories and addressing matrices to store the compressed information, we treat the total number of images as a memory storage budget. When

| | I/C | DC [10] | DSA [11] | KIP (NN) [12] | CAFE* [46] | TM [15] | DM [14] | Ours |
|---|---|---|---|---|---|---|---|---|
| | 1 | 91.7±0.5 | 88.7±0.6 | 90.1±0.1 | 93.1±0.3 | - | 89.7±0.6 | **98.7±0.7** |
| MNIST [41] | 10 | 97.4±0.2 | 97.8±0.1 | 97.5±0.0 | 97.5±0.1 | - | 97.5±0.1 | **99.3±0.5** |
| | 50 | 98.8±0.2 | 99.2±0.1 | 98.3±0.1 | 98.9±0.2 | - | 98.6±0.1 | **99.4±0.4** |
| | 1 | 70.5±0.6 | 70.6±0.6 | 73.5±0.5 | 77.1±0.9 | - | - | **88.5±0.1** |
| F-MNIST [42] | 10 | 82.3±0.4 | 84.6±0.3 | 86.8±0.1 | 83.0±0.3 | - | - | **90.0±0.7** |
| | 50 | 83.6±0.4 | 88.7±0.2 | 88.0±0.1 | 88.2±0.3 | - | - | **91.2±0.3** |
| | 1 | 31.2±1.4 | 27.5±1.4 | 57.3±0.1 | 42.9±3.0 | - | - | **87.3±0.1** |
| SVHN [43] | 10 | 76.1±0.6 | 79.2±0.5 | 75.0±0.1 | 77.9±0.6 | - | - | **89.1±0.2** |
| | 50 | 82.3±0.3 | 84.4±0.4 | 80.5±0.1 | 82.3±0.4 | - | - | **89.5±0.2** |
| | 1 | 28.3±0.5 | 28.8±0.7 | 49.9±0.2 | 31.6±0.8 | 46.3±0.8 | 26.0±0.8 | **66.4±0.4** |
| CIFAR10 [44] | 10 | 44.9±0.5 | 52.1±0.5 | 62.7±0.3 | 50.9±0.5 | 65.3±0.7 | 48.9±0.6 | **71.2±0.4** |
| | 50 | 53.9±0.5 | 60.6±0.5 | 68.6±0.2 | 62.3±0.4 | 71.6±0.2 | 63.0±0.4 | **73.6±0.5** |
| CIFAR100 [44] | 1 | 12.8±0.3 | 13.9±0.3 | 15.7±0.2 | 14.0±0.3 | 24.3±0.3 | 11.4±0.3 | **34.0±0.4** |
| | 10 | 25.2±0.3 | 32.3±0.3 | 28.3±0.1 | 31.5±0.2 | 40.1±0.4 | 29.7±0.3 | **42.9±0.7** |
| TinyImageNet [45] | 1 | - | - | - | - | 8.8±0.3 | 3.9±0.2 | **16.0±0.7** |

Table 1: We compare our method with previous works on ConvNet recovered accuracy. Our algorithm consistently outperforms all previous methods and achieves state-of-the-art. *Note that we selected the best results from baseline model variants. I/C is images per class (storage budget eqn. 4).

comparing with $N$ images for $C$ classes, we ensure

$$\text{size(bases)} + \text{size(addressing matrices)} \approx NC\text{size(image)} \qquad (4)$$

where $\text{size}(\cdot)$ is the total size of a tensor, assuming the numbers are stored as floats. For a given number of bases, we calculate the corresponding maximum number of addressing matrices allowed using eqn. 4 and use the integer lowerbound as the final value.

**Model details.** We use bases with spatial resolution downsampled by a factor of 2 in both height and width from the standard image size based on datasets. All models are trained for 50k iterations with SGD optimizer. For the inner loop optimization, we set the momentum rate as 0.9, and use 150 steps for small memory budgets 1 and 10, and 200 steps for budget 50. The number of bases is selected on the held-out validation set (10% of training set), an example is shown in figure 2, where different numbers of bases and addressing matrices (calculated with eqn. 4) have impacts on accuracies; more details in the appendix.

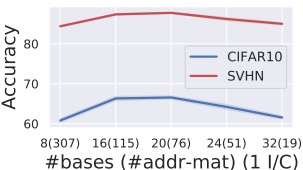

Figure 2: Validation set

**Result 1: state-of-the-art accuracy.** We compare our model with previous methods: Dataset Condensation (DC) [10], Differentiable Siamese Augmentation (DSA) [11], Kernel Inducing Points (KIP) [12], Distribution Matching (DM) [14], Aligning Features (CAFE) [46] and Trajectory Matching (TM) [15]. The results are summarized in table 1. Following previous methods [11, 12, 15], we adopt simple data augmentations and preprocessing: flip and rotation on CIFAR10 and CIFAR100 datasets, and ZCA on SVHN, CIFAR10 and CIFAR100. As shown in table 1, our model consistently outperforms previous methods, especially under high compression rate cases where only 1 image is allowed per class. For example, we achieve 87.3% and 66.4% on SVHN and CIFAR10 with 1 image per class, outperforming prior arts by 30% and 16.5% under the same storage budget, respectively, and even beat the performance of previous methods using 10 images per class.

**Analysis: information sharing across classes.** The core observation in our method is that a common representation can enable information sharing across classes and reduce redundancies. To verify this, we calculate the average coefficients $\bar{v} = \frac{1}{r} \sum_{i=0}^{r-1} y^T A_i$ for each class $y$ and visualize the cosine similarities of $\bar{v}$ from two classes. The visualizations are shown in fig. 3. Higher cosine similarity scores indicate that two classes are utilizing similar bases components in the memories to produce synthetic images. For example, in CIFAR100 (right one of figure 3), classes maple, oak, palm, pine and willow trees have strong sharing, while lawn mower and rocket are distinct from each other. Similar patterns can be found in CIFAR10 dataset, shown in the left one of figure 3.

**Result 2: back-propagation through time is a strong baseline.** In figure 4 and table 2, we show that a vanilla BPTT variant is already a strong baseline which outperforms previous single-step gradient methods [10] by 40.4% on SVHN and 20.3% on SVHN and CIFAR10 under 1 image per class. Note that the performance on SVHN has doubled the accuracy 31.2% obtained using single-step gradient matching methods [10]. In the vanilla BPTT variant, no downsampling (ds) or

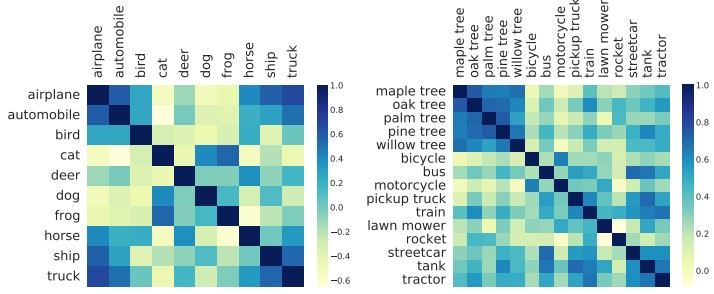
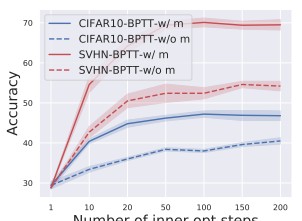

Figure 3: Similarity matrices of learned addressing coefficients for the CIFAR10 dataset (left) and a subset of CIFAR100 classes (right).

Figure 4: Analysis on BPTT steps and momentums.

| | I/C | Single-step GM | Ours$^{\text{BPTT}}$ | Ours$^{\text{BPTT+ds}}$ | Ours$^{\text{Full w/o Aug.}}$ | Ours$^{\text{Full}}$ |
|---|---|---|---|---|---|---|
| CIFAR10 | 1 | 28.8±0.7 | 49.1±0.6 | 55.2±0.5 | 64.2±0.6 | **66.4±0.4** |
| | 10 | 52.1±0.5 | 62.4±0.4 | 65.9±0.4 | 70.9±0.4 | **71.2±0.4** |
| | 50 | 60.6±0.5 | 70.5±0.4 | 71.1±0.5 | 72.1±0.5 | **73.8±0.4** |
| CIFAR100 | 1 | 13.9±0.3 | 21.3±0.6 | 25.9±0.4 | 33.5±0.2 | **34.0±0.4** |
| | 10 | 32.3±0.3 | 34.7±0.5 | 36.5±0.4 | 40.6±0.3 | **42.9±0.7** |

Table 2: Ablation studies of every component and comparison with single-step gradient matching [10]. ds: downsampling. Aug.: data augmentation.

memory addressing formulation is used. We also analyze the effects of *long unrolls* and *momentum terms* on vanilla BPTT in figure 4. It is observed that on both short inner loops (10 steps) and long ones (100 steps), adding momentum terms can consistently lead to a strong performance boost, e.g. 7.0% and 9.2% on CIFAR10. Using longer inner loop trajectories can also increase the recovered accuracy, e.g. 18.2% and 42.3% on CIFAR10 and SVHN, respectively, compared to 1 step cases.

**Ablation studies.** To further analyze the effects of different components in our algorithm, we perform ablation studies on CIFAR10 and CIFAR100. Besides the vanilla BPTT, the ablation results of components (downsampling, augmentation and memory addressing formulation) are summarized in table 2. We show that downsampling can indeed reduce spatial redundancies (e.g. improve results from 49.1% to 55.2% on CIFAR10 with 1 image per class), and memory addressing formulation can further increase the recovered accuracy (from 55.2% to 64.2% on CIFAR10 with 1 image per class). It is also shown that our model is quite robust to the ablation of data augmentation, which has a small effect (1-2%) on the results. The resulting algorithm is a *simple and effective* framework that uses memory addressing formulation and BPTT with long unrolls to distill datasets.

**Cross-architecture generalization.** Our memories and addressing matrices are also generalizable across various architectures. We test our algorithm on ConvNet, ResNet12 and AlexNet for training and testing. The results are summarized in the appendix, section **??**, table **??**.

## 5.2 Continual learning

One of the key usages of memories is to prevent forgetting when a model continually learns through tasks. In this section, we evaluate our algorithm on the standard continual learning benchmarks and show that, due to the strong performance, a simple "compress-then-recall" method with our model can already rival with previous state-of-the-arts with complex designs.

**Efficient lifelong learning.** Following [47], we work with the problem where all tasks are streamed in mini-batches and learned in a *single pass*. A learner is allowed to be equipped with a small memory buffer. The data samples after seen will not be available unless stored in the buffer. We use a mini-batch size of 10 to stream the data, following previous works [28, 24].

**Evaluation.** The learner's performance after learning on the task stream is commonly evaluated under two metrics: retained accuracy (RA) and backward-transfer and interference (BTI). RA is the average accuracy of the final trained model on all tasks, and BTI measures the performance difference between after it was learned and after the full training process. Note that our algorithm does not perform actual learner training on the data streams and BTI is not applicable.

| | Rotations | | Permutations | | MANY | | CIFAR-100 | |
|---|---|---|---|---|---|---|---|---|
| | RA↑ | BTI↓ | RA↑ | BTI↓ | RA↑ | BTI↓ | RA↑ | BTI↓ |
| ONLINE | $53.38^{\pm1.53}$ | -5.44 | $55.42^{\pm0.65}$ | -13.76 | $32.62^{\pm0.43}$ | -19.06 | $32.62^{\pm0.43}$ | -19.06 |
| EWC [48] | $57.96^{\pm1.33}$ | -20.42 | $62.32^{\pm1.34}$ | -13.32 | $33.10^{\pm0.14}$ | -18.50 | - | - |
| GEM [24] | $67.38^{\pm1.75}$ | -18.02 | $55.42^{\pm1.10}$ | -24.42 | $39.50^{\pm0.62}$ | -17.50 | $48.27^{\pm1.10}$ | -13.7 |
| MER [17] | $77.42^{\pm0.78}$ | -5.60 | $73.46^{\pm0.45}$ | -9.96 | $51.00^{\pm0.54}$ | -13.57 | $51.38^{\pm1.05}$ | -12.83 |
| La-M [28] | $77.42^{\pm0.65}$ | -8.64 | $74.34^{\pm0.67}$ | -7.60 | $50.43^{\pm0.21}$ | -10.00 | $61.18^{\pm1.44}$ | -9.00 |
| sp-La [30] | $77.77^{\pm0.58}$ | -8.16 | $76.88^{\pm0.72}$ | -8.39 | $50.81^{\pm0.79}$ | -13.73 | - | - |
| Ours | $\mathbf{80.32}^{\pm0.28}$ | N/A | $\mathbf{78.48}^{\pm0.76}$ | N/A | $\mathbf{74.07}^{\pm0.51}$ | N/A | $\mathbf{62.58}^{\pm1.1}$ | N/A |

Table 3: We show that "compress-then-recall" is a strong baseline that outperforms previous methods on four continual learning benchmarks. Baseline numbers are from [28] or obtained from public official repos.

**Benchmarks.** We evaluate our method on three tasks widely used in previous Continual Learning works. MNIST Rotations [24] contains 20 tasks with 1,000 samples in each. Every task consists of images rotated by a fixed angle from 0 to 180 degrees. MNIST Permutations [48] has 20 tasks, and each task contains 1,000 images generated through shuffling the image pixels by a fixed permutation. MANY Permutations [17] is a longer variant with 100 tasks in total and 200 samples in each. Incremental CIFAR-100 [29, 24] splits the CIFAR100 dataset into 20 5-way classification tasks as the task stream for learning.

**Our model.** Based on our distillation method, we adopt a simple framework to perform continual learning: "compress then recall". During the training phase, we do not perform learning on neural networks, instead, the dataset of each task is distilled to memories and the paired addressing matrices. During test phase, we simply fetch the corresponding memories and addressing matrices for each task, and train a new model from scratch to perform classification. *Memory buffer designs.* When a new task starts, we use the full remaining memory buffer to store the samples and perform distillation with both buffer samples and streamed samples. After a task ends, the distilled memories and addressing matrices are stored in the buffer, taking $1/T$ of the space, where $T$ is the total number of tasks. Namely, the buffer size keeps shrinking when more compressed representation of tasks is stored. Note that we compare our model with previous methods under the *exact same memory sizes for fair comparisons*. See the appendix for more details on model and memory designs.

**Results.** We show that this simple method is already a strong baseline that outperforms prior arts on four benchmarks, summarized in table 3. Our method is compared with: Online, EWC [48], GEM [24], MER [17], C-MAML [28], La-MAML [28] and Sparse-LaMAML [30]. For example, we can obtain a 23% boost on MNIST MANY benchmark: from 50.81% to 74.07%.

We further compare our model with previous works Kernel Continual Learning [39] and Stable SGD [37] following their settings, where each task in MNIST Rotation and MNIST Permutation contains 60,000 samples instead of 1,000 samples. Our model achieves $87.3^{\pm0.92}$ and $88.3^{\pm0.58}$ on Permutated MNIST and Rotated MNIST under their setting, outperforming both KCL ($85.5^{\pm0.78}$ and $81.8^{\pm0.60}$) and Stable SGD ($80.1^{\pm0.51}$ and $70.8^{\pm0.78}$). Interestingly, our results also are higher than the multitask upperbound ($86.5^{\pm0.21}$ and $87.3^{\pm0.47}$), potentially due to that there is task interference in joint training, which can be naturally avoided in our method.

## 5.3 Synthesizing new classifiers after learning

If we want to memorize the past, what is the benefit of storing the compressed representation rather than a trained model? In this section, we show that our compressed representation can enable flexible synthesis of new classifiers after the learning. Specifically, we demonstrate extrapolating between tasks to train new models, and performing memory recall with images instead of labels, showing the generalizability of our framework on other query forms.

### 5.3.1 Extrapolating between tasks

In the real world, tasks often do not come together and a learner, therefore, cannot observe all tasks at once. In current machine learning paradigms, when models are separately trained for disjoint tasks, it has difficulty extrapolating between tasks to build new classifiers. This is different from human learning. We show that storing our compressed representation enables a learner to extrapolate and synthesize new classifiers after learning separately on each task. Specifically, we separate CIFAR100

into 20 disjoint 5-way classification tasks as training tasks. For testing, we select classes that are not seen together during training by randomly choosing $k$ tasks and picking 1 class from each selected task. We use $k = 2$ and $k = 5$ to construct 2-way and 5-way classification tasks, and sample 1,000 tasks each for evaluation. To train our models, we independently distill the datasets for 20 training tasks into corresponding memories. For each testing task, the class labels are used as queries to recall the synthetic data from the corresponding memories. The recalled data for each label, although not seen together during training, are used for re-training a $k$-way classifier from scratch. We find that the compressed data can indeed train classifiers on new combinations, for example, we can achieve 72.53%±8.74 on 2-way classification, and 46.54%±6.42 on 5-way classification, with 1 image per class storage budget. The upperbound with the full real dataset is 92.23%±4.76 and 82.72%±4.29.

### 5.3.2 Dataset Distillation extension – recall the past with images

We extend the standard setting to recall the past with images: when the label information and task scopes are missing, but a few visual observations can be made, we would like to build classifiers based on the visual data. For example, when we see a bear image and a deer image, but cannot recall the exact word or category, can we recall the memories with images and build a classifier? This is possible with our problem formulation, where the forms of queries are not constrained to labels and we can *distill a dataset to memories addressable by images*.

We formulate the problem as follows. Formally, after observing a training dataset $\mathcal{D}_{tr}$ with $\mathcal{Y} = \{0, ..., C\text{-}1\}$, we would like to flexibly build classifiers for a subtask $\mathcal{Y}_g \subset \mathcal{Y}$ based on visual observations $\mathcal{X}_g$ from $\mathcal{Y}_g$, when the actual information of $\mathcal{Y}_g$ is unknown. We work with 1-shot and 5-shot observation cases. As a baseline, we build a nearest neighbor classifier, which is pretrained on $\mathcal{D}_{tr}$ and takes features of few-shot data

| Methods | 1 shot | 5 shot |
|---|---|---|
| Nearest neighbor | 48.55 | 61.72 |
| classify-then-recall | 50.58 | 58.46 |
| image addressing | **55.74** | **71.20** |

Table 4: Few-shot perf. recovery.

to classify test images. As a model variant, we could also "classify-then-recall", using a classifier trained on $\mathcal{D}_{tr}$ to map the image shots into labels and turning into the standard setup. Benefiting from the general design of our formulation, we show that a model can directly perform "image addressing", where a feature network can provide query vectors $\boldsymbol{y}$ in fig.1. The feature network, memories and addressing matrices can be jointly trained on $\mathcal{D}_{tr}$. We evaluate the above models and baselines on CIFAR100 and summarize the results in table 4. As shown in the table, our model is not only able to successfully perform the *continuous query addressing* with image feature vectors, but also outperforms two strong baselines on constructing new classifiers. More analysis is in the appendix.

## 6 Conclusion and limitations

In this paper, we propose a framework that distills a large dataset into compact addressable memories. This framework introduces several benefits, including removing the linear growth contraints on the compressed data size, allowing more general queries besides categorical labels, and most importantly, achieving high compression rate with strong re-training performance, outperforming previous state-of-the-arts in dataset distillation. We also demonstrate a "compress-then-recall" method using our framework, leading to new state-of-the-arts in continual learning on four datasets. Our full model has potential limitations on the costly inner optimization loop, which might be time-consuming on larger models or datasets. This limitation might be solved by combining the memory formulation with a different learning framework. One potential societal concern with dataset distillation in general is that the distilled dataset may not contain the full diversity of the original data distribution, causing the retrained classifier to perform especially poorly on minority populations; our method arguably takes a step towards mitigating that concern through improving the retrained accuracy.

## 7 Acknowledgements

This material is based upon work supported by the National Science Foundation under Grants No. 2107048 and 2112562. Any opinions, findings, and conclusions or recommendations expressed in this material are those of the author(s) and do not necessarily reflect the views of the National Science Foundation. We would also like to thank Vishvak Murahari, Sunny Cui, Ruth Fong, Vikram Ramaswamy, and Zeyu Wang for discussions.

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
