# A   Experiment setups

In this section, we provide detailed experimental setups for all the tasks discussed in the main paper. Specifically, we will explain the datasets, architectures and implementation details for all tasks.

## A.1   Dataset Distillation

**Datasets.** Our models are tested on six standard dataset distillation benchmarks:

- MNIST contains 10 classes with 60,000 writing digit images as the training set and 10,000 images as the test set. The images are gray-scale with a shape of $28 \times 28$ and associated with a label from 10 classes (digit 0-9).

- FashionMNIST is a dataset with clothing and shoe images and consists of a training set with size 60,000 and a test set with size 10,000. Each image is $28 \times 28$ in gray scale, and has a label from 10 classes.

- SVHN street digit images where each image has a shape of $32 \times 32 \times 3$. The dataset contains 73257 images for training and 26032 images for testing. We use the cropped SVHN where the center of the image indicates the number and the rest is background. Each image is categorized into 10 classes (digits 0-9).

- CIFAR10 is a dataset consisting of $32 \times 32$ RGB images and has 10 classes in total: airplane, automobile, bird, cat, deer, dog, frog, horse, ship, and truck. Each class contains 5,000 images for training and 1,000 images for testing, leading to 50,000 images for training and 10,000 images for testing in total.

- CIFAR100 contains 60,000 images in total from 100 classes. For every class, 500 images are used for training and 100 images are used in testing. The 100 classes are associated with 20 superclasses, where each superclass contains 5 classes at a finer level.

- TinyImageNet is a downscaled subset of ImageNet, with 200 classes. The dataset contains images of shape 64x64, a training set with 100,000 images and a testing set with 10,000 images.

**Architectures.** We mainly work with a three-layer convolutional neural network, denoted as "ConvNet", which contains convolutional layers with $3 \times 3$ filters, followed by ReLU activation function and InstanceNorm. The network has 128 hidden dimensions and uses an average pooling layer with $2 \times 2$ kernel size after every Instancenorm operation. We also test our models on ResNet-12 with 64, 128, 256, 512 hidden dimensions in each block. The ResNet-12 architecture is slightly modified by replacing BatchNorm with InstanceNorm, and removing the final average pooling layer. We find using the full spatial information in the final layer is important for distillation. Both ConvNet and ResNet-12 are standard architectures for few-shot learning benchmarks.

**Implementation details.** We use one 24-GB GPU for each experiment run. For all our models, we use a SGD optimizer with learning rate 0.1 and momentum rate 0.5. Every model is trained for 50,000 iterations. For both the inner loop optimization and evaluation, we use learning rate 0.01 and momentum rate 0.9. For random initialization of addressing matrices and bases, we use Kaiming uniform initialization. To select the number of bases for each setting, we randomly sample 10% of training set as the validation set. Data augmentations with rotation and flip are applied on CIFAR10 and CIFAR100 datasets. ZCA preprocessing is used on CIFAR10, CIFAR100 and SVHN datasets. No ZCA preprocessing or data augmentations are used on MNIST and FashionMNIST datasets.

## A.2   Continual learning

**Datasets.** We use six datasets to evaluate our models. The details are summarized in table 1.

**Architectures and implementation details.** For all MNIST-based datasets, we use a multi-layer perceptron (MLP) with 256 hidden units. Following La-MAML, we use the ConvNet architecture with 160 hidden dimensions. All experiments are run on a 24-GB GPU, using a SGD optimizer with 0.1 learning rate and 0.5 momentum rate. The inner loop optimization learning rate is set as 0.01 with momentum rate 0.9. During the testing phase, the re-training phase uses the same setups as the inner loop optimization. We use data samples stored in the memory buffer for minibatch replay to perform compressing, summarized in table 1.

| Dataset | #Tasks | Batch size | #Samples/task | Total mem size | #Bases | #Replay |
|---------|--------|------------|---------------|----------------|--------|---------|
| Rotations | 20 | 10 | 1000 | 200 | 24(ds) | 4 |
| Permutations | 20 | 10 | 1000 | 200 | 8 | 20 |
| MANY | 100 | 10 | 1000 | 1000 | 8 | 20 |
| CIFAR100 | 20 | 10 | 2250 | 200 | 24(ds) | 4 |
| Rotations* | 20 | 10 | 60000 | 200 | 24(ds) | 2 |
| Permutations* | 20 | 10 | 60000 | 200 | 8 | 2 |

Table 1: The details on six benchmarks used in the experiments: MNIST Rotations (Rotations), MNIST Permutations (Permutations), MANY Permutations (MANY), Incremental CIFAR100 (CI-FAR100), MNIST Rotations with 60,000 data samples (Rotations*), MNIST Permutations with 60,000 data samples (Permutations*). Note that works compare under different benchmarks, we follow the settings and compare our model with La-MAML on Rotations, Permutations, MANY, and CIFAR100, and compare with Kernel Continual Learning on Rotations* and Permutations*. #Samples per task is specified for training. (ds) indicates using downsampled bases.

## A.3    New classifier synthesis

**Datasets and setups.** For experiments on both extrapolating between tasks and recall with images, we use CIFAR100 as the dataset. *For task extrapolation experiments*, we split CIFAR100 into 20 5-way classification tasks for training, and use 2-way and 5-way classification for testing. The 2-way and 5-way tasks during testing are obtained through randomly selecting 2 or 5 training tasks and then randomly sampling 1 class from each selected task. This ensures that every pairs of classes in a testing task have not been used together for training. *For recall with images experiments*, we use all classes in CIFAR100 for training, and use 20 5-way classification tasks in testing. During evaluation on a 5-way classification task, we sample 1 or 5 images per class (depends on 1-shot or 5-shot), and use the sampled images for recall. The sampled images are from test set, i.e. we would like to use testing images to perform recall and build a new classifier.

**Architectures and implementation details.** In the task extrapolation experiments, since our models are performing dataset distillation, we use the exact same hyperparameters and architectures as dataset distillation tasks in Sec. A.1. For recall with images, we use 64 bases and 16 addressing matrices (i.e. each query can generate 16 synthetic images) in our model. For baselines, we pretrain the feature backbone for nearest neighbor classifier and the classifier in "classify-then-recall" for 100 epochs on CIFAR100, using SGD optimizer with 0.01 learning rate and 0.9 momentum rate. The visual observations (image shots) we used are from the test set.

## B    Additional results and discussion

### B.1    Dataset Distillation

**Back-propagation through time as a strong baseline.** Besides the main ablation study results on CIFAR10 and CIFAR100, table 3 provides the results for the benchmarks. As shown in the table, back-propagation through time is indeed a strong baseline that consistently outperforms the single-step gradient matching method, and downsampling can reduce spatial redundancies and improve the compression rate, leading to a higher recovery performance.

| | 1 image/class | | | 10 images/class | | |
|---|---|---|---|---|---|---|
| | AlexNet | ResNet-12 | ConvNet | AlexNet | ResNet-12 | ConvNet |
| AlexNet | 58.5±0.5 | 53.6±0.6 | 57.3±0.6 | 65.6±0.5 | 60.2±0.6 | 63.7±0.6 |
| ResNet12 | 53.2±0.8 | 58.5±0.5 | 57.0±0.3 | 62.3±0.9 | 67.8±0.3 | 65.2±0.6 |
| ConvNet | 50.5±1.3 | 55.9±0.6 | 66.4±0.4 | 63.8±0.8 | 67.5±0.4 | 71.2±0.4 |

Table 2: Cross architecture generalization under various pixel/image storage budgets.

|        | I/C | Single-step GM | Ours$^{\text{BPTT}}$ | Ours$^{\text{BPTT+ds}}$ | Ours$^{\text{Full w/o Aug.}}$ | Ours$^{\text{Full}}$ |
|--------|-----|----------------|----------------------|-------------------------|-------------------------------|----------------------|
| MNIST  | 1   | 91.7±0.5       | 95.2±0.3             | 98.2±0.1                | -                             | **98.7±0.7**         |
|        | 10  | 97.4±0.2       | 98.8±0.1             | 98.9±0.1                | -                             | **99.3±0.5**         |
|        | 50  | 98.8±0.2       | 99.2±0.1             | 99.4±0.1                | -                             | **99.4±0.4**         |
| F-MNIST| 1   | 70.5±0.6       | 83.9±0.4             | 86.7±0.3                | -                             | **88.5±0.1**         |
|        | 10  | 82.3±0.4       | 89.1±0.2             | 89.1±0.1                | -                             | **90.0±0.7**         |
|        | 50  | 83.6±0.4       | 90.4±0.1             | 90.7±0.1                | -                             | **91.2±0.3**         |
| SVHN   | 1   | 31.2±1.4       | 71.6±0.8             | 80.1±0.5                | -                             | **87.3±0.1**         |
|        | 10  | 76.1±0.6       | 83.1±0.3             | 86.2±0.2                | -                             | **89.1±0.2**         |
|        | 50  | 82.3±0.3       | 86.5±0.2             | 88.8±0.2                | -                             | **89.5±0.2**         |
| CIFAR10| 1   | 28.3±0.5       | 49.1±0.6             | 55.2±0.5                | 64.2±0.6                      | **66.4±0.4**         |
|        | 10  | 44.9±0.5       | 62.4±0.4             | 65.9±0.4                | 70.9±0.4                      | **71.2±0.4**         |
|        | 50  | 53.9±0.5       | 70.5±0.4             | 71.1±0.5                | 72.1±0.5                      | **73.6±0.5**         |
| CIFAR100| 1  | 12.8±0.3       | 21.3±0.6             | 25.9±0.4                | 33.5±0.2                      | **34.0±0.4**         |
|        | 10  | 25.2±0.3       | 34.7±0.5             | 36.5±0.4                | 40.6±0.3                      | **42.9±0.7**         |

Table 3: Full ablation studies on model variants and comparison with single-step gradient matching baseline. No augmentations are used on MNIST, FashionMNIST and SVHN.

**Transfer across architectures.** To show that our compressed memories are generalizable across architectures, we also test the training on ResNet-12. Specifically, we learn the memories and addressing matrices on ConvNet and ResNet-12, and test them on ResNet-12 and ConvNet, respectively. Results are summarized in table 2. We use 10 images per class as the storage budget on CIFAR10. Each row is the architecture that our method trains on, and each column is the generalization performance. The learned compressed representation is quite robust across ConvNet and ResNet-12.

**Choice of # bases.** To select the number of bases for each experiment, we evaluate the performance on a separate validation set, which is 10% random samples of the training set. The results on the validation set are shown in fig. 1. We select the number of bases that leads to the highest performance on the validation set for the full training set distillation.

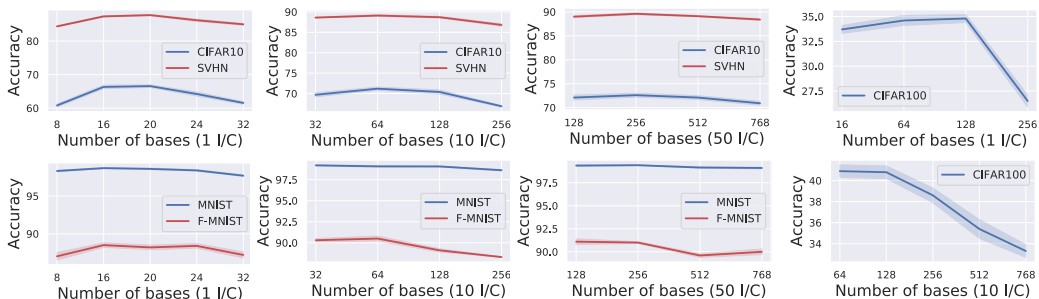

Figure 1: Number of bases v.s. retrain accuracy on validation set. I/C: images per class.

**Further ablations on momentum terms.** How is the momentum term exactly affecting the backpropagation through time process? We analyze the performance of baseline BPTT algorithms on three cases: no-momentum, forward-only momentum, and full momentum. No-momentum uses BPTT without momentum terms. Forward-only momentum uses the momentum term only in the forward BPTT, but blocks the gradients on the momentum term in the backward pass (except for the gradients on the current time step weights) to remove the "bridging effect" of momentum term across multiple steps. Full momentum is our full model. All the experiments are performed on CIFAR10 with 200 inner optimization steps.

| I/C | no-momentum | forward-only momentum | full momentum |
|-----|-------------|-----------------------|---------------|
| 1   | 40.5±0.8    | 45.6±0.7              | 49.1±0.6      |
| 10  | 50.0±0.5    | 57.4±0.3              | 62.4±0.4      |

Table 4: Further analysis of momentum terms of BPTT on CIFAR10 dataset.

**Adam optimizer for inner loop.** We also experimented with using Adam optimizer to optimize the synthetic data, instead of using stochastic gradient descent with momentum. Empirically, we found that Adam optimizer leads to certain instability of gradients (e.g., magnitude) on the inner optimization steps when using the same learning rate magnitude and perfers smaller ones such as 1e-4. The end results are similar to the SGD algorithms.

## B.2 Continual learning

**Memory designs in "compress-then-recall".** We follow the Reservoir sampling strategy to store samples in the memory buffer. When learning through the tasks, our algorithm utilizes all the currently available memory buffer storage space to store samples. After the learning on one task is finished, the algorithm saves the compressed representation to the memory buffer, taking $1/T$ the buffer where $T$ is the total number of tasks, and clear the storage space which stores the real samples for the current task. This strategy makes sure that the compression algorithm has enough samples to replay, resulting in $1 - (t-1)/T$ of the storage to use, where $t \in \{1, ..., T\}$ is the current task index.

## B.3 New classifier synthesis

**Designs of "image addressing" model.** Since our formulation allows flexible query forms, we use an extra ConvNet to take the visual observations (images) as input and treat the output feature vectors as queries. The feature vector queries are used for vector matrix product with addressing matrices to compute coefficients for combining bases. To train the addressing model: For every training iteration, we randomly subsample a subset of classes from all classes and pick 1 or 5 images (depends on 1-shot or 5-shot), and use the recalled synthetic datasets with the feature vectors of the images to perform inner loop optimizations. The generalization loss is computed using other image-label pairs from the subset classes (the same as standard dataset distillation training). The ConvNet (feature extractor), bases and image matrices are jointly trained.

**Discussion of "image addressing" results.** We compare the "image addressing" model with two strong baselines: nearest neighbor classifiers and "classify-then-recall" method. It's interesting to see that, having the ability to access the dataset-level information (even compressed) can often lead to better performance when building a new classifier, while nearest neighbor classifiers can only utilize image shots to serve as limited information for classification. Note that the "classify-then-recall" method is also a strong baseline, but can suffer from the classification errors on test images, leading to less robust recall. The direct usage of feature vectors from image shots can provide a continuous space and potentially lead to more robust behaviours in the addressing and recall processes.

## C Visualization and analysis

**Coefficients similarity map.** We show the full matrix of cosine similarities on the coefficients that combine the bases from all 100 classes in CIFAR100, as shown in fig. 2. The order of classes on x and y axis is organized by superclasses. Every 5 classes is under a common superclass on the axis. As shown by the matrix, we can clearly see that the classes under the same superclass often have significant similarities, indicating strong sharings when combining bases. For example, categories bridge, castle and house share similar bases; baby, girl, man and woman also share similar bases, while crab and tulips use very different coefficients to perform addressing.

**Visualization on bases.** In figure 3, we visualize the learned 64 bases on CIFAR100. The bases contain various colors, shapes and textures, and are used to be combined with coefficients generated from queries and addressing matrices.

## D More visualization comparisons

In this section, we further compare the visualization of our methods under various settings.

## D.1 Same amount of generated images

We visualize the synthetic images from the baseline method BPTT and from our proposed memory addressing parameterization. For BPTT, we use 100 image per class as the budgets, and for our

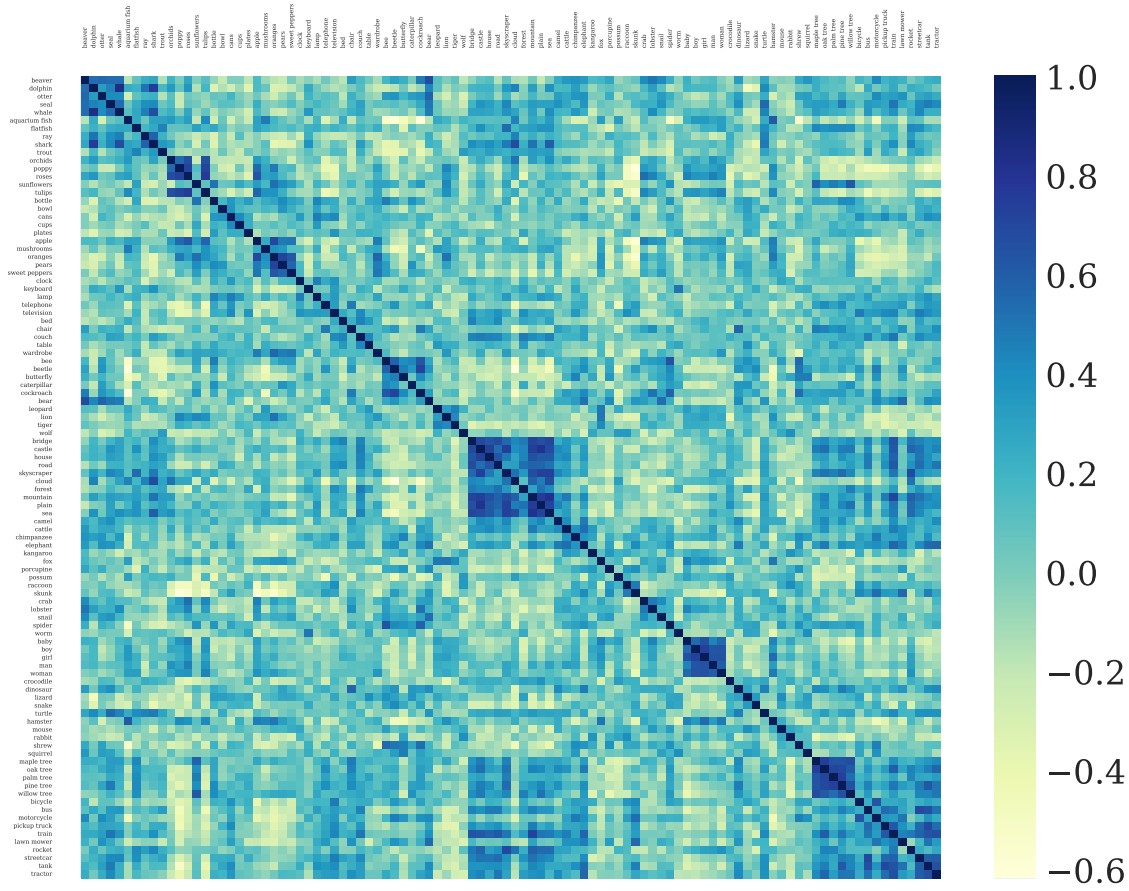

Figure 2: Full coefficient cosine similarity matrix on CIFAR100. Zoom in to view the details. Classes are ordered with superclasses. On the x and y axis, in order, every 5 classes belongs to a common superclass.

method, we use 10 images per class and 43 bases to generate approximately the same amount of recalled images (99). The synthetic images are visualized in figures below. To easily compare with the vanilla version of BPTT, we do not use downsampling in either BPTT or the memory addressing formulation.

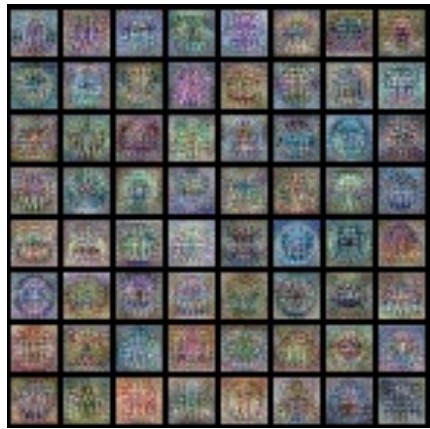

Figure 3: CIFAR100 learned 64 bases.

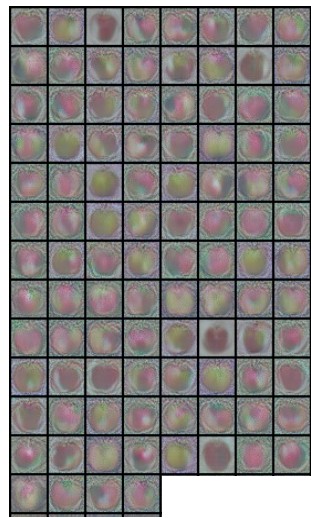

Method: BPTT with standard parameterization

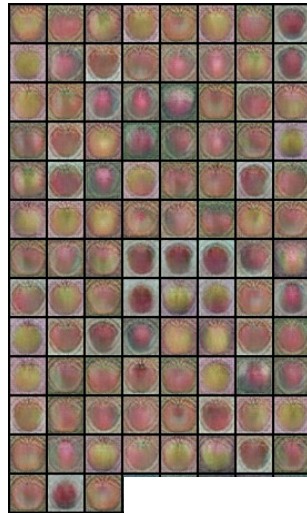

Method: BPTT with memories and addressing matrices

Figure 4: Recalled synthetic images for class apple.

## D.2 Various image per class budgets

We compare the visualizations of synthetic images under various image per class (I/C) budgets. Similar to previous section D.1, we use bases with the same shape, and compare the results under 2, 10 and 50 I/Cs. The corresponding number of bases are 8, 43 and 215. The visualization results are summarized in figure 8, figure 9 and figure 10,

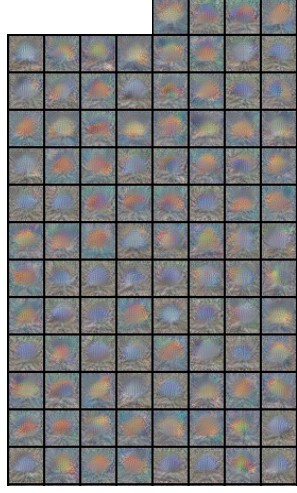

Method:  BPTT with standard parameterization    Method: BPTT with memories and addressing matrices

Figure 5: Recalled synthetic images for class aquarium fish.

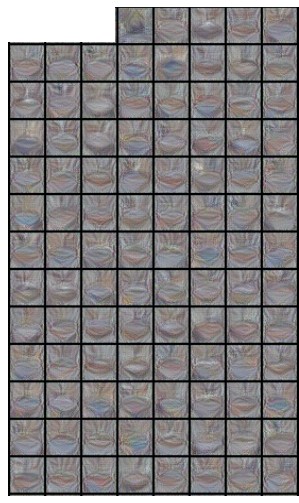

Method:  BPTT with standard parameterization    Method: BPTT with memories and addressing matrices

Figure 6: Recalled synthetic images for class bed.

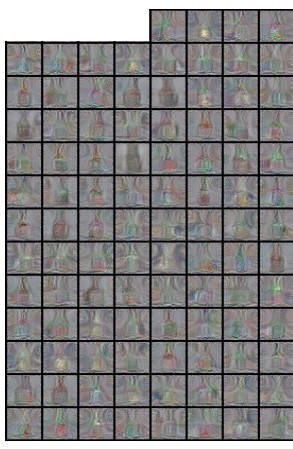

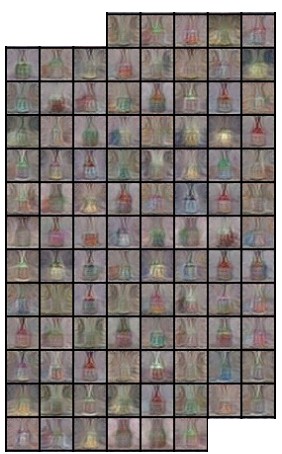

Method: BPTT with standard parameterization          Method: BPTT with memories and addressing matrices

Figure 7: Recalled synthetic images for class bottle.

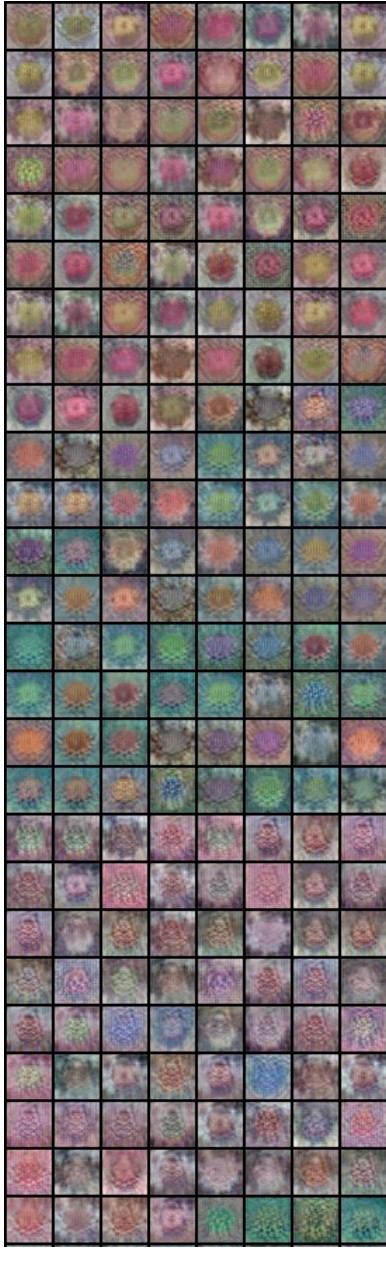

Figure 8: Recalled synthetic images for classes apple, aquarium fish, and baby under 2 I/C with 8 bases.

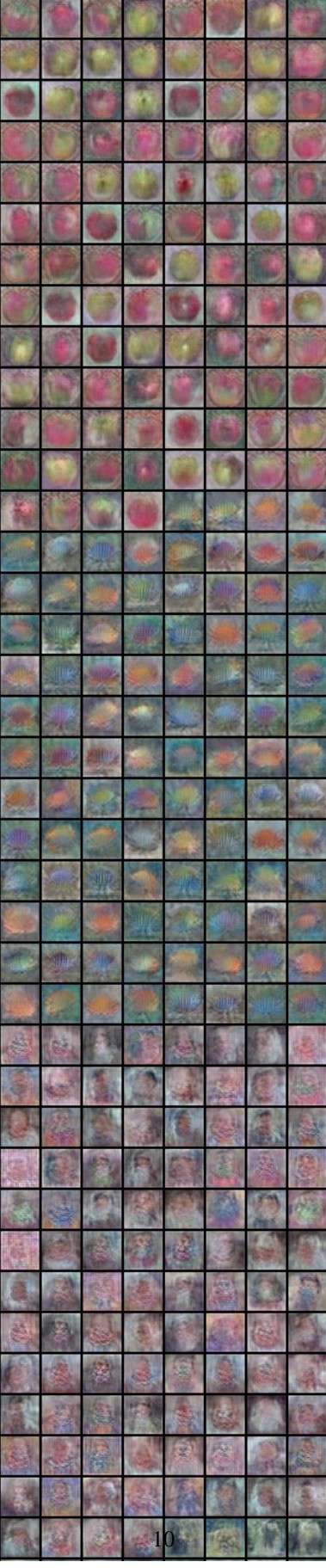

10

Figure 9: Recalled synthetic images for classes apple, aquarium fish, and baby under 10 I/C with 43 bases.

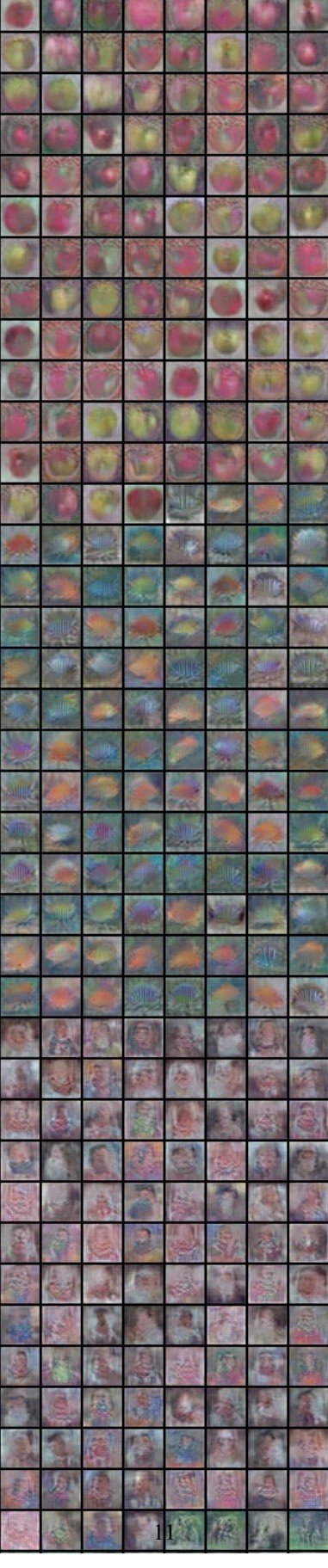

Figure 10: Recalled synthetic images for classes apple, aquarium fish, and baby under 50 I/C with 215 bases.