# OpenReview forum: "Remember the Past: Distilling Datasets into Addressable Memories for Neural Networks"
_NeurIPS.cc/2022/Conference — NeurIPS 2022 Accept_

### Official Review · Reviewer_3HqT · 2022-07-03

**Rating:** 6
**Confidence:** 5
**Soundness:** 3 good
**Presentation:** 2 fair
**Contribution:** 3 good

**Summary:**

The paper proposes to compress the critical information of a large dataset into compact addressable memories. This method achieves better results than the previous dataset distillation/condensation methods.

**Questions:**

1. Algorithm 1. Lines 8 - 13 are the detailed optimization algorithm for $\theta$. In line 15, there is the optimization algorithm for $\phi$. What are the details of this optimization algorithm?
2. There is also a research direction that generative models are used to condense a dataset. This direction is closely related to this work and should be discussed in the paper.
3. In previous data distillation works, the number of inner-loop optimization steps is small. The major reason is that the memory consumption and execution time are quite large with a large trajectory. Could the authors discuss how they address this issue?
4. There is a trade-off between time and space. If we prefer a higher compression rate, we have to pay the cost of extra computation (e.g., the computation for decompression). Could the authors discuss this trade-off and make a comparison with previous work?
5. Instead of reporting the performance given a specified compression rate, it is also important that what is the compression rate if we can fully recover the original performance. Namely, what is the size of the smallest dataset that can be trained to achieve the full dataset performance?
6. The authors use the notation $M=${$b_0, ..., b_{K-1}$}. It can be simplified as $M \in R^{K \times d}$.
7. Is it possible to compress the condensed dataset directly? For instance, the previous methods can generate a condensed dataset. Could we apply several lossless compression on them? It is better for the authors to discuss it.

**Limitations:**

As a derived method of the dataset distillation framework, the proposed method inherits the limitations of the original method on the large models and large datasets. The authors also mention it in the last section.

**Strengths And Weaknesses:**

Strengths.
1. The results are much better than previous methods.
2. The method is based on a reasonable assumption that there are redundancies in the previous condensed dataset. The method achieves a higher compression rate by reducing redundancies.
3. The paper is well organized and developed.

Weaknesses.
1. The experiments are only on small datasets. The size of the dataset $N$ and the dimension of input $d$ are small. The authors should report results on large datasets. Specifically, in [15], Tiny ImageNet and ImageNet subsets are used in the experiments.
2. The experiments on large models instead of small models are needed. Does the method depend on a specific neural architecture? The authors may investigate other neural networks.
3. Other missing experiments are mentioned in the questions below.
4. The writing is fair. Several typos and errors are listed below.

* Line 45. significanly -> significantly
* Line 50. lead to -> leads to
* Line 63. There has been -> There have been
* Line 64. criterions -> criteria
* Line 66. It emphasize on -> It emphasizes
* Line 70. gaussian processes -> Gaussian processes
* Line 79. dillema -> dilemma
* Lines 92, 100, 151. e.g. -> e.g.,
* Line 99. the number of synthetic data sample -> the number of synthetic data samples
* Line 115. seprately -> separately
* Line 121. outperforms -> outperform
* Line 146. defines -> define
* Line 170. having to enumerating -> having to enumerate
* Equation 3. J(\phi) -> min J(\phi)
* Line 179. discuss in detail about the learning framework -> discuss in detail the learning framework
* Line 212. there are strong evidence -> there is strong evidence
* Line 213. there are information re-using -> there is information re-using
* Line 223. with shape 28 × 28 -> with a shape of 28 × 28
* Line 228. are color image 228 dataset -> are color image 228 datasets
* Line 263. distinct to each other -> distinct from each other
* Line 292. 200 sample -> 200 samples
* Line 308. pervious -> previous

---

> ### Author Response · Authors · 2022-08-02
> **Author response to reviewer 3HqT**
>
> We would like to thank the reviewer for the detailed and valuable comments. A new revision is submitted with an updated draft. We address the specific questions below.
>
> **Q. small datasets and more diverse models**
>
> Thanks for the suggestions. We are running experiments with larger images. Note that BPTT takes longer times on higher resolution data. We will include our findings (whether positive or negative) with thorough explorations in the final version, along with the computational cost of these experiments. Current still very early results show that our algorithm can achieve 9.8% on 1 I/C TinyImageNet (compared to 8.8% using trajectory matching [15]).
>
>
> We have verified our algorithm on larger models and cross-architecture performance:
>
> 1 I/C:
>
> |          | AlexNet    | ResNet12   | ConvNet    |
> |----------|------------|------------|------------|
> | AlexNet  | 58.53(0.5) | 53.63(0.6) | 57.32(0.6) |
> | ResNet12 | 53.21(0.8) | 58.49(0.5) | 57.01(0.3) |
> | ConvNet  | 50.50(1.3) | 55.86(0.6) | 66.4(0.4)  |
>
> 10 I/C:
>
> |          | AlexNet   | ResNet12  | ConvNet   |
> |----------|-----------|-----------|-----------|
> | AlexNet  | 65.6(0.5) | 60.2(0.6) | 63.7(0.6) |
> | ResNet12 | 62.3(0.9) | 67.8(0.3) | 65.2(0.6) |
> | ConvNet  | 63.8(0.8) | 67.5(0.4) | 71.2(0.4) |
>
> **Q. writing update**
>
> We have cleared the typos and updated a new draft in the revision. Thank you for the very detailed comments!!
>
> **Q. line 15 optimization algorithm**
>
> We follow previous works and use SGD with momentum 0.5 and learning rate 0.01 as the optimizer.
>
> **Q. Generative models**
>
> Thanks for the suggestion. We will add the discussion to the paper.
>
> **Q. Small number of steps in BPTT and memory constraints**
>
> Note that one critical reason hinging the previous work is the lack of momentum terms. For the memory constraint, the GPU memory of our BPTT is constant in time. To solve the memory problem, we manually coded the BPTT back-propagation process, instead of relying on the computation graph and autograd in Pytorch. Our manual implementation leads to constant memory (does not linear grow with time), but needs 1.3x time comparing to autograd with full computation graphs (the models need to commute between CPU and GPU).
>
> **Q. trade-off between time and space**
>
> Since the current dataset distillation works are not lossless compression, the decompression results vary across settings, leading to difficulties in the trade-off discussion (compared to lossless compression where the target is always the original data).
>
> The main bottleneck is now still how to distill information more effectively in to the data. For example, with a better distillation algorithm, more informative content can be distilled in one image per class and the benefits come for free - the model can be re-trained using the same amount of distilled data and achieve higher performance. In our algorithm, since we use bases-coefficients decomposition, more coefficients can lead to higher re-training time. But this can be solved using larger batch sizes.
>
> **Q. full recovering rate**
>
> Thanks for the suggestion. Note that in the current Dataset Distillation works [10,13,14,15], achieving full recovering rate is still an open problem (e.g., DM[14] sacrifices accuracies for training speed and has pushed it to 95% relative recovering rate on CIFAR10, but still hasn't achieved 100%. We will discuss it in our main paper.
>
> **Q. notation**
>
> We will update the notation in the text and equations in the final version.
>
> **Q. lossless compression**
>
> Thanks for the interesting suggestion. The lossless compression can be applied to both distilled data from previous methods and our memory representations. The major difference between lossless compression and the line of dataset distillation works is information prioritization - whether the compressed data affects the downstream decision making process. Comparing to applying a standard compression, it would be interesting instead to consider differentiating through a standard compression algorithm for dataset distillation and learning the bits that affect the downstream optimization process. We leave it as the future work.
>
> **Q. limitations**
>
> See **Beyond performance** and **BPTT limitations** in  [this response](https://openreview.net/forum?id=RYZyj_wwgfa&noteId=UhI-_IA2rLl4).

---

### Official Review · Reviewer_AR9N · 2022-07-06

**Rating:** 6
**Confidence:** 3
**Soundness:** 3 good
**Presentation:** 4 excellent
**Contribution:** 4 excellent

**Summary:**

Based on the observation that classes can share concepts and representations, the authors propose a new Dataset Distillation method that trains a learnable memory shared across classes. The objective is to learn basic concepts that can be shared and then, through a function A, learn to combine these concepts to generate examples given a class y_i. To learn the method, the authors use a bi-level optimization process. The inner loop verified the generalization properties of the current memory, and the outer loop updated the memory and attention function. Additional to the method, the authors propose a modification in the momentum factor during the inner loop that helps improve performance. This method achieves good results in various Dataset Distillation benchmarks and good results in Continual Learning scenarios.

**Questions:**

Q.1 In several works, the attention masks are learned independently of the model/memory weights, so that interference between the learning processes is avoided. Does it make sense to learn M and A separately? In some ways, this experiment seems helpful to ensure that the memory acquires basic concepts, and not that function A selects useful random vectors that can achieve a goal, like finding the lottery ticket from a memory.

Q.1.2 Perhaps the previous question is related to what happens in Fig. 2. For some classes, there is an intuitive and testable relationship of similarity. However, this similarity is not seen in other intuitively similar classes (cats and dogs). Is there any explanation for what is happening?

Q.3 One of the problems with using long paths in the inner loop is the amount of memory used to store the entire associated computation graph. Although there are techniques to mitigate this, it is not mentioned. Can this proposal be explained in more detail? Has this problem been encountered?

Q.4 It is unclear how the method is used in the Continual Learning environment. A single-pass context is mentioned. However, how can the proposed method be learned in just one data pass? Did the simple-pass apply only to the test adaptation? Or is the model capable of learning to compress information in a single epoch?

**Limitations:**

The authors mention a limitation that is very relevant and I share

**Strengths And Weaknesses:**

Strengths:
- S.1 The observation that concepts are shared across tasks is very intuitive. This observation, as the authors note, helps compress data so that it does not grow linearly as the number of classes increases, and may also lead to finding better concepts for classes that the model has not considered before.not seen.
- S.2 The paper is well written, placing much emphasis on the different contributions that the authors propose. The different concepts and notations used are well explained.
- S.3 Using the bi-level optimization strategy helps generate weights that can support the problem of continuous labels.
- S.4 The results show that the proposed method achieves outstanding results. The experiments that were carried out demonstrate the contributions proposed by the authors.

Weaknesses:
- W.1 The reason for adding the momentum update is not clear. Nor how it is done. Is there any intuition behind this contribution?
- W.2 Is there a reason why line 12 of Algorithm 1 does not use the opt function? It is used in equation 3, but not in the algorithm, which can cause confusion.
- W.3 It is known that benchmarks of the dataset distillation problem are standard and normally use only small images. However, many of the previous methods have problems when the size of the images increases. It would be interesting to confirm the performance of this method on larger images. Alternatively, add it to the limitations.
- W.4 There is no comparison of the algorithm's time or the number of iterations. Adding more iterations to the inner loop must come at a high cost.
- W.5 Typo: Line 110, the abbreviation BPTT is used before defining

---

> ### Author Response · Authors · 2022-08-02
> **Author response to reviewer AR9N**
>
> We would like to thank the reviewer for the detailed and valuable comments. We address the specific questions below.
>
> **Q: momentum term**
>
> (Also explained in other responses)
>
> Since we are using the BPTT process, the end positions of the inner optimization are critical and decide what information can be backpropagated through the optimization process. In our experiments, we find that the momentum term can help on:
>
> (1) Producing the optimized parameters that better summarize the distilled datasets (with the smoothing effects). For example, when we perform the forward pass using momentum, but block the gradients on the momentum term in the backward pass (except for the gradients on the current time step weights), the performance is still higher than the no-momentum version, indicating that the end position (when obtained with momentum terms) produces more informative gradients for training.
>
> (2) Momentum term is the summation of decayed forward gradients on multiple time steps (alg. line 11). The gradients on the outer loop loss (alg. line 15) can be backpropagated via the momentum through multiple previous time steps, potentially mitigating the gradient vanishing issue. The rationale for using momentum for BPTT (in the different context of hyperparameter tuning) is also discussed in [1].
>
> How the backpropagation on momentum is performed: we refer the reviewer to paper [2] alg. 2 for details. The BPTT forward process is unfolded in alg. box lines 11-12. The back-propagation process on BPTT is a standard chain-rule derivation that intuitively uses the momentum term as a “bridge” to propagate gradients directly to previous steps. We will also add explicit details to the main paper.
>
> **Q: alg. 1 line 12**
>
> We explicitly write out the update rule in line 12 to highlight the details of momentum usage. It can be also wrapped as an opt operation.
>
> **Q: larger images**
>
> Thanks for the suggestion. We are running experiments with larger images. Note that BPTT takes longer times on higher resolution data. We will include our findings (whether positive or negative) with thorough explorations in the final version, along with the computational cost of these experiments. Current still very early results show that our method can achieve 9.8% on 1 I/C TinyImageNet (compared to 8.8% using trajectory matching [15]). Besides larger images, we also validated the results with more model architectures in response to reviewer 3HqT.
>
> **Q: algorithm's time and the solution of memory constraints**
>
> We show the training time v.s. achieved accuracy in the following table. We will add more discussions on the training time.
>
> | mins | 46 | 120 | 600 |
> |------------|------|------|------|
> | accuracy   | 0.40 | 0.43 | 0.48 |
>
> | mins | 285 | 720 | 2350 |
> |------------|------|------|------|
> | accuracy   | 0.57 | 0.61 | 0.65 |
>
> The GPU memory of our BPTT is constant in time. To solve the memory problem, we manually write the BPTT back-propagation process, instead of relying on the computation graph and autograd in Pytorch. Our manual implementation leads to constant GPU memory (does not linearly grow with time), but needs 1.3x time compared to autograd with full computation graphs  (the models need to commute between CPU and GPU).
>
> **Q: line 110**
>
> We have updated the draft.
>
> **Q: separate learning of M and A**
>
> Thanks for the interesting suggestion! Separate training of memories and attentions can indeed lead to benefits with Sigmoid or Softmax attentions in feature learning. The lottery ticket situation can probably happen with sigmoid attentions and with large enough redundancies. Note that our model uses the linear matrix and does not constrain the values to [0,1]. Also, due to it is a compression problem, there is not much redundancy in the representation or budget. Better training of memories and matrices (with various regularities or attention forms) itself is indeed a quite interesting direction. We leave it for future works.
>
> **Q: similarities in figure 2**
>
> The cat and dog similarity in figure 2 is still higher than most other pairs, such as cat and automobile. The eventual similarity is determined by the effectiveness in model optimization, e.g. the model might decide a medium-level similarity leads to better performance.
>
> **Q: continual learning in one pass (epoch)**
>
> Yes, our model can distill the datasets and achieve good performance in a single epoch. We use memory buffer and multi-replay to increase the sample size and optimization steps. But even without the Reservoir memory buffer to store more real data samples, our performance is still pretty good (e.g., 77.6% on RotationMNIST, compared to the current 80.32%).
>
> **Limitation**
>
> See **BPTT limitation** and **Beyond performance** in [this response](https://openreview.net/forum?id=RYZyj_wwgfa&noteId=UhI-_IA2rLl4).
>
> [1] Gradient-based Hyperparameter Optimization through Reversible Learning. ICML 2015
>
> [2] Forward and Reverse Gradient-Based Hyperparameter Optimization. ICML 2017

---

### Official Review · Reviewer_fXia · 2022-07-12

**Rating:** 7
**Confidence:** 4
**Soundness:** 3 good
**Presentation:** 3 good
**Contribution:** 3 good

**Summary:**

This paper formulates the dataset distillation as a memory addressing process where the synthetic datasets can be generated through an addressing matrix based on a learned memory representation. The proposed method improves the compression rate of distilled data by considering the class similarity and improving the BPTT framework. The proposed method is evaluated on five image classification datasets and applied to continual learning and few-shot learning.

**Questions:**

Table 1
- The comparison to previous methods is not perfectly fair, and "I/C" is misleading. Due to the parameterization, the proposed method can generate more synthetic data than previous methods. Therefore, the proposed method is expected to achieve higher accuracy than previous methods. Thus, the number of addressing matrices (or the exact number of synthesis examples) needs to be shown somewhere in the table. I want to see whether more distilled data can fully explain the improvement or if the distilled images also get better quality. Appendix Figures 4 and 5 show that the recalled images lack diversity. I am surprised that the method achieves such a good performance.

Table 2
- It may be better to clarify the abbreviation (i.e., ds, Aug) in the caption.

Figure 3
- You mention that unrolling the trajectories long enough (e.g., 200 steps) can improve performance (Line 202). Could you show a complete picture in Figure 3 up to 200 steps? Due to the long unroll of the BPTT, I would also like to see a plot of test accuracy versus training time and the memory requirement of the proposed method. How does it compare to other baseline methods?

Figure 4
- I am curious about the trade-off between the number of bases and addressing matrices given a fixed memory budget. What is the general rule to scale these two numbers? I want some plot where the x-axis is the memory budge, and the y-axis is the ratio between these two numbers.

Results 2
- I can see that long unrolls will be very important, but it is unclear why momentum can be so significant? What's the insight behind that?

Line 301
- "the buffer size keeps shrinking when more compressed representation of tasks is stored": does it imply that the later distilled data tend to have a low quality?

Misc
- What would happen if you use a soft label for query (as a data augmentation) during test time? Will the performance get improved?
- It will be interesting to see the visualization of the interpolation between two different classes. Does the "half dog half ship" image make sense?
- I want to see some visualization of the images generated by BPTT. How does it compare to the previous method?

**Limitations:**

Due to the long unroll of the BPTT, the training efficiency and memory requirement can be the biggest concern when scaling this approach to complex datasets or larger models.

**Strengths And Weaknesses:**

This paper is well-motivated and focuses on a significant problem (the parameterization of the distilled data) in dataset distillation. The proposed method is novel, and the results are encouraging.
- Originality: The proposed method proposes a new way to parameterize the distilled data and opens up a new application for few-shot learning.
- Quality: This paper is technically sound, and most of the arguments are well supported. The evaluation is not perfectly fair, and additional study is needed to understand the method better.
- Clarity: This paper is easy to follow and mostly well-written, with minor flaws.
- Significance: The proposed method achieves state-of-the-art performance when a fixed memory budget is given. The addressable memory parameterization can also be applied to other dataset distillation methods. This paper also shows that dataset distillation can benefit continual learning and few-shot learning.

---

> ### Author Response · Authors · 2022-08-02
> **Author response to reviewer fXia (1/2)**
>
> We would like to thank the reviewer for the detailed and valuable comments. A new revision is submitted with the updated draft. We address the specific questions below.
>
> **Q: I/C Comparison**
>
> We updated the revision in the caption on I/C (we also emphasized it in the main figure and equation 4). Note that the parameterization and forms of data are also a fundamental question for Dataset Distillation. An effective parameterization can facilitate the understanding of what critical information affects the training of models. I/C is actually _measuring and reflecting this perspective and property of algorithms_. A recent work [1] from ICML’22 also discusses the parameterization of dataset distillation. We will add further clarifications and potentially use pixels per class following [1] for better clarity.
>
> The number of generated images does not necessarily correspond to the performance. Having more coefficients under the same budget does not always lead to a strong advantage over a smaller number of coefficients. For example, CIFAR10 and SVHN with 307 coefficients underperform fewer coefficients (115, 76, 51, or 19) in figure 4. We think it is the decomposition on representation that leads to good performance — the parameterization is an effective way to capture the principal components and the variations in datasets.
>
> **Q: visualization vs performance**
>
> Indeed there are some empirical observations in each work on how the appearance of images might correlates with performance. However, we believe that there are still no formal connections built (e.g., through extensive experiments, human studies or theories) between what kind of appearances of images can necessarily lead to a strong performance. The visualization more potentially shows the property of the algorithm class. For example, with single-step gradient matching methods, the distilled data can potentially contain more high frequency and local information from the data. For BPTT which emphasizes the endpoint position of parameters, the contours of objects and how contours might vary (even in small scales) seem to be more critical.
>
> **Q: Abbreviation**
>
> We will update this in the final version.
>
> **Q: Figure 3**
>
> We have updated the figure in the revision.
>
> **Q: training time vs accuracy, memory requirement**
>
> We show the performance progresses along with time. The following tables are under CIFAR10 1 I/C budget, without memory representation (table 1) and with memory representation (table 2). Our algorithm can achieve high performance relatively fast.
>
> Table 1:
>
> | mins | 46 | 120 | 600 |
> |------------|------|------|------|
> | accuracy   | 0.40 | 0.43 | 0.48 |
>
> Table 2:
>
> | mins | 285 | 720 | 2350 |
> |------------|------|------|------|
> | accuracy   | 0.57 | 0.61 | 0.65 |
>
> To solve the memory problem, we manually write and code the BPTT back-propagation process, instead of relying on the computation graph and autograd in Pytorch. Our manual implementation leads to constant GPU memory (does not linearly grow with time) but needs 1.3x time compared to autograd with full computation graphs (the models need to commute between CPU and GPU).
>
> Compared to the single-step gradient matching algorithm, BPTT is slower. The multi-step trajectory matching [15] can also distill in shorter time, but requires pretraining of a model database. We discuss more details of BPTT in **BPTT limitation**.
>
> **Q: plot of memory budget and bases ratios**
>
> Following the reviewer's suggestion, we compute the ratios of the number of bases (#basis) and number of mats (\#mats) under different budgets (number of images per class), shown in table 3. Besides the suggested one, we also computed the ratios of the total number of dimensions, shown in table 4. We will include this analysis in the paper.
>
> Table 3:
>
> | \#basis/\#mats | 1    | 10   | 50   |
> |--------------|------|------|------|
> | MNIST        | 0.55 | 0.14 | 1.92 |
> | FashionMNIST | 0.55 | 0.63 | 0.45 |
> | SVHN         | 0.26 | 0.16 | 0.49 |
> | CIFAR10      | 0.26 | 0.16 | 0.49 |
> | CIFAR100     | 8.0  | 0.55 |   -  |
>
> Table 4:
>
> | \#basis-dim/\#mat-dim | 1    | 10   | 50   |
> |---------------------|------|------|------|
> | MNIST               | 0.68 | 0.09 | 0.15 |
> | FashionMNIST        | 0.68 | 0.19 | 0.07 |
> | SVHN                | 1.01 | 0.19 | 0.15 |
> | CIFAR10             | 1.01 | 0.19 | 0.15 |
> | CIFAR100            | 0.48 | 0.03 |   -  |
>
> **Q: trade-off between the number of bases and addressing matrices**
>
> The trade-off is shown in figure 4. In our experiments, we first compute all possible value pairs for (number of bases, number of addressing matrices), select the middle ones with a reasonable amount of matrices, and choose the one with the highest performance on the validation set. Note that the performance is in general quite similar among different \#bases-\#matrices as long as it is not close to the edge cases (too few bases to capture principal components or too few coefficients to capture variations).

---

> > ### Author Response · Authors · 2022-08-02
> > **Author response to reviewer fXia (2/2)**
> >
> > **Q: momentum term**
> >
> > Since we are using the BPTT process, the end positions of the inner optimization are critical and decide what information can be backpropagated through the optimization process. In our experiments, we find that the momentum term can help on:
> >
> > (1) Producing the optimized parameters that better summarize the distilled datasets (with the smoothing effects). For example, when we perform the forward pass using momentum, but block the gradients on the momentum term in the backward pass (except for the gradients on the current time step weights), the performance is still higher than the no-momentum version, indicating that the end position (when obtained with momentum terms) produces more informative gradients for training.
> >
> > (2) Momentum term is the summation of decayed forward gradients on multiple time steps (algorithm line 11). The gradients on the outer loop loss (algorithm line 15) can be backpropagated via the momentum through multiple previous time steps, potentially mitigating the gradient vanishing issue. The rationale for using momentum for BPTT (in the different context of hyperparameter tuning) is also discussed in [2].
> >
> > **Q: buffer size in Continual Learning**
> >
> > Having more buffer size to store real data for distillation can indeed lead to higher performance on early tasks. This is quite common in continual learning with the Reservoir sampling strategy (e.g., in [3]). But our algorithm's performance drop is relatively small. For example, on MNIST-Rotation and CIFAR100, we show the accuracies of every 4 tasks and the last 4 tasks:
> >
> > MNIST Rotation
> >
> > every 4: 81.37, 80.59, 81.12, 81.83, 80.18
> >
> > last 4:  78.13, 80.18, 75.36, 76.65
> >
> > CIFAR100
> >
> > every 4: 62.2, 76.6, 57.2, 71.0, 53.4
> >
> > last 4:  53.4, 63.0, 62.4, 53.0
> >
> > **Misc:**
> >
> > Using soft labels as queries during testing is a very interesting idea. This is essentially interpolating between classes with different weights. Another possible variation could be interpolating the coefficients (addressing outputs) within each class or label as augmentations. Note that, currently, the coefficients are un-normalized, leading to potentially different scales at each dimension. We leave this for future works and thank the reviewer for this suggestion.
> >
> > Visualization of BPTT (no memory representation): very similar to the appendix ones. We found the BPTT in general produces images that emphasize quite a lot on the contours of objects and tend to capture less on specific textures.
> >
> >
> > **Q: BPTT limitation**
> >
> > As mentioned in section 6, BPTT is relatively slow as it requires solving the inner optimization process. However, note that our proposed representation can be flexibly applied to other distillation frameworks, this major contribution is orthogonal to previous works.
> >
> > There is a large bulk of works on tackling or approximating the inner process of BPTT, ranging from implicit differentiation to approximation methods (e.g., truncation, real-time approximate learning). We believe providing observation on the full unapproximated BPTT has shown the potential of this algorithm and the bi-level optimization procedures, and can lead to more explorations in this direction.
> >
> > ***Beyond performance**
> >
> > Besides the benchmarks with fixed settings, our algorithm introduces several important properties:
> >
> > (1) Flexible budgets. We can handle various target budgets, such as difficult-to-balance ones (e.g., 150 images over 100 classes), or float budgets (e.g., 3.5 I/C). This advantage of our algorithm is critical in the broader and practical usage of dataset distillations.
> >
> > (2) Not linearly grow with the number of classes. This benefit can be critical in future works which target datasets with large number of classes. For example, in ImageNet21k, it is almost unavoidable to adopt other representations, since even 1 I/C will lead to 21,000 images in the standard way of parameterizing the distilled data.
> >
> > (3) Addressable memories open the directions to other tasks and data modalities. A key advantage of our algorithm is that it allows continuous queries as “labels” to address the memories and obtain distilled images. This in general makes it possible for the distillation of image datasets where queries are from other modalities, such as audios, language, or even image themselves (image-image pairs in datasets).
> >
> > **References**
> >
> > [1] Dataset Condensation via Efficient Synthetic-Data Parameterization, ICML 2022
> >
> > [2] Forward and Reverse Gradient-Based Hyperparameter Optimization. ICML 2017
> >
> > [3] On Tiny Episodic Memories in Continual Learning, Arxiv 2019

---

> > > ### Comment · Reviewer_fXia · 2022-08-07
> > > **Response to Authors**
> > >
> > > Thanks for the response. I appreciate this paper's idea of the new parameterization of the distilled data, and I still have the following questions.
> > >
> > > Q: I/C Comparison
> > > - I would like to know more about the generated images' quality. For example, I want to see the performance with 100 coefficients compared to the baseline methods that distill 100 images. I know that it is not fair in terms of storage. It is a good way to tell whether the images have better quality or the performance improvement is mainly due to the new parameterization that allows you to generate more images.
> > >
> > > Q: visualization vs. performance
> > > - Why do you think "contour" is more critical for the BPTT type of algorithm? In other words, why does gradient matching learn high frequency and local information, and why does BPTT capture the contours? Besides, could you provide more synthetic image visualizations when you increase the I/C budget for several classes which you think are representative? I want to understand the order of each class's "principle component" and whether there is a trend that the synthetic images become more diverse and diverse in what sense.
> > >
> > > Q: Abbreviation
> > > - Why not do it in the current version? I feel like it hurts the reading experience.
> > >
> > > Q: plot of memory budget and bases ratios
> > > - Does Table 3 mean K/r and Table 4 mean K_d/(r_K*d_y)? From Table 3, it seems that there is a U-shape behavior where we want a high ratio with a low or high memory budget and a relatively low ratio when we have a medium budget. What is your interpretation of this observation? And, what is the best practice or intuition to choose the proper ratio when considering a new dataset with a certain memory budget? What can we do besides performing a hyperparameter sweep on the validation set?
> > >
> > > Q: momentum term
> > > - Could you provide any quantitative results to see the effect of each hypothesis? If (1) turns out to be more critical, then does it mean the main reason why we want to use more BPTT steps is to find a better-optimized parameter? Then, would an algorithm that converges faster (e.g., Adam) give us better performance?
> > >
> > > Q: buffer size in Continual Learning
> > > - I'm confused about what you mean by every 4 tasks and the last 4 tasks. Could you clarify a bit?
> > >
> > > Misc
> > > - It will be great also to include a visualization of BPTT, even if it looks similar. Otherwise, the readers will not know that information.
> > >
> > > Beyond performance
> > > - I agree with the contributions to the dataset distillation community.

---

> > > > ### Author Response · Authors · 2022-08-09
> > > > **Response to additional questions (2/2)**
> > > >
> > > > **Q: plot of memory budget and bases ratios**
> > > >
> > > > Yes, table 3 counts the number of bases and addressing mats, and table 4 counts the total number of dimensions to store the bases and addressing mats. Regarding the U-shape behavior, it’s a great observation. Our interpretation is that, when the storage budgets are high, the algorithm would need more bases to be expressive enough to generate data that capture more detail-level information in the dataset that is useful for optimization, leading to higher ratios.
> > > >
> > > > In general, there can be several factors to consider when working with a new dataset: (1) the complexity of the dataset — intuitively, when the dataset is simple, there might not be a strong need for too many bases, for example, the models can work well on MNIST without having too many bases; (2) the sharing among classes —  when there is a strong sharing among classes (through observation of the dataset), the number of bases can be reduced; (3) expressiveness vs variations trade-off — with more bases, the set of reusable components are more powerful, with more coefficient matrices, the more variations of composing the principal components can be captured in the distilled data. Based on the practical usage, it might be worth considering which part needs more capacity depending on the datasets or tasks.
> > > >
> > > > This is a great question. Thanks for diving into the details of our algorithm!
> > > >
> > > > **Q: momentum term**
> > > >
> > > > For sure, below are the performance of no-momentum, forward-momentum only, and full momentum. No-momentum uses BPTT without momentum terms. Forward-only momentum uses the momentum term only in the forward BPTT, but blocks the gradients on the momentum term in the backward pass (except for the gradients on the current time step weights). Full momentum is our full model. All the experiments are performed on CIFAR10 with 200 inner optimization steps.
> > > >
> > > > | Accuracy              | 1 I/C      | 10 I/C     |
> > > > |-----------------------|------------|------------|
> > > > | no-momentum           | 40.5 (0.8) | 51.0 (0.5) |
> > > > | forward-only momentum | 45.6 (0.7) | 57.4 (0.3) |
> > > > | full momentum         | 49.1 (0.6) | 62.4 (0.4) |
> > > >
> > > >
> > > > We did try Adam and found that it leads to unstableness when being used as the inner loop optimizer. Specifically, the gradients on the distilled data are not stable, leading to difficulties in the learning process.
> > > >
> > > > **buffer size in Continual Learning**
> > > >
> > > > Sorry for the confusion, we instead provide the accuracies of all tasks under a random seed. The reported accuracies are the retained performance after observing the full task sequence.
> > > >
> > > > MNIST Rotation
> > > >
> > > > Performance (accuracies) of all tasks (1-20):
> > > >
> > > > 82.28, 81.37, 82.44, 84.11, 81.66, 80.59, 82.32, 81.48, 83.41, 81.12, 80.78, 82.5, 80.67, 81.83, 79.17, 84.03, 78.13, 80.18, 75.36, 76.65
> > > >
> > > > CIFAR100
> > > >
> > > > Performance (accuracies) of all tasks (1-20):
> > > >
> > > > 62.2, 63.0, 61.2, 55.8, 76.6, 53.6, 68.2, 55.6, 57.2, 69.6, 69.6, 62.8, 71.0, 60.2, 63.2, 56.0, 53.4, 63.0, 62.4, 53.0
> > > >
> > > > **Misc**
> > > >
> > > > Yes, we agree. the visualizations are included in the appendix.
> > > >
> > > > **Beyond performance**
> > > >
> > > > Thank you. We appreciate that the reviewer also agrees with the contributions beyond the performance.
> > > >
> > > > [1] George Cazenavette, Tongzhou Wang, Antonio Torralba, Alexei A Efros, and Jun-Yan Zhu.  Dataset distillation by matching training trajectories. CVPR’2022
> > > >
> > > > [2] Deep Leakage from Gradients Ligeng Zhu, Zhijian Liu, Song Han, NeurIPS’2019
> > > >
> > > > [3] Dataset Condensation via Efficient Synthetic-Data Parameterization, ICML'2022

---

> > > > ### Author Response · Authors · 2022-08-09
> > > > **Response to additional questions (1/2)**
> > > >
> > > > Thank you for the follow-up questions! We have made the corresponding updates in the main paper and appendix. Let us know if you have other questions. We are happy to elaborate and discuss.
> > > >
> > > > **Q. Visualization**
> > > >
> > > > We have updated the appendix in section D “More visualization comparisons”. Two settings are added as suggested. In the first setting, we compare the visualizations of synthetic images on dataset CIFAR100 from four classes: apple, aquarium fish, bed, and bottle. The vanilla BPTT baseline has 100 images per class and the memory addressing algorithm (ours) has 10 images per class budget with 43 bases, leading to 99 recalled images per class. In the second setting, we show the recalled images under 2, 10, and 50 I/C budgets, with 8, 43, and 215 bases respectively. As we are comparing with the vanilla BPTT, we also used bases with the original resolution to be fair. We’ll add all the suggested visualization and analysis to the final version as well.
> > > >
> > > > **Q: visualization vs. performance**
> > > >
> > > > Thanks for the questions. Intuitively, BPTT emphasizes only the end position of parameters in the space. This potentially means that there are no strong constraints on gradient trajectory — the loss can be minimized as long as it reaches the correct final position. The _single-step gradient matching_ method imitates the gradient trajectories from the real datasets (which is a very sensible choice) and indicates that it tries to replicate the whole searching process from the optimizer (e.g., in [3]). Note that, during the search process, there can be gradient steps that only try to escape from some local regions, potentially leading to short-range behaviors and local information to distill. The recent work MTT/TM[1] also discussed the problem and nicely dealt with it and uses multi-step gradient matching to directly match the start and end positions of fragments in teacher trajectories. Another observation is from Deep Gradient Leakage [2]: through using the single-step gradients, it’s possible to approximately recover the whole image in a batch, while the goal of dataset distillation is to extract only the useful parts. This also indicates that there is extra information besides the core parts which are solely useful for optimization.
> > > >
> > > > Note that we think both BPTT and gradient matching methods are great frameworks and have strong potential in Dataset Distillation. It would be nice to see the efforts on either side to push the algorithms forward and generate stronger distilled data or broaden the field to other tasks or applications.
> > > >
> > > > **Q: Abbreviation**
> > > >
> > > > Thanks, we agree and have updated the draft in both the caption and the paragraph. We will explain more details about table 2 and update the notations in both captions and paragraphs, once have more space.

---

> ### Comment · Reviewer_fXia · 2022-08-09
> **Thanks for the response.**
>
> Thanks for the detailed answers. Since there is little time left for the author-reviewer discussion, I will increase my score to 7 (accept). But, I hope the authors can address all my remaining questions and add the new experiments to the appendix.
>
> Q: I/C Comparison
> - Thanks for the visualization. Have you ever trained a model on the 99 recalled images per class? I want to know more about how it compares with the vanilla BPTT baseline with 100 images per class.
>
> Q: plot of memory budget and bases ratios
> - From your interpretation, it seems that the number of bases dominates the expressive power of the distilled data. However, the fixed bases seem to have very limited expressive power compared to a set of learnable parameters. It would be interesting to see what will happen if you further parameterize the bases similar to the attention layer in Transformer. There is no need to run any experiments, and I hope the authors can explore more in future work.
>
> Q: momentum term
> - Thanks for the results. It seems that backpropagating through the momentum term really helps the performance. As for Adam, if you use it in a forward-only manner, why will it cause unstableness? Have you tuned the learning rate (I guess it needs at least 10x smaller)? However, if you mean the instability when backpropagating through Adam, you might want to add a small epsilon term to the second momentum term. Besides, it would be interesting to see the performance of forward-only Adam.
>
> Q: buffer size in Continual Learning
> - I'm confused about what you mean by every 4 tasks and the last 4 tasks. Could you clarify a bit?
>
> Q: related work
> - Here is a recent paper [1] that uses 1-step TBPTT, which seems to have good training and memory efficiency. Their distilled images also seem real and natural. What is your thought on that? Do you think replacing the BPTT component with their method can yield a better result?
>
> [1] Zhou, Yongchao, Ehsan Nezhadarya, and Jimmy Ba. "Dataset Distillation using Neural Feature Regression."

---

> > ### Author Response · Authors · 2022-08-09
> > **Thanks for the questions and suggestions!**
> >
> > Thank you for all the additional feedback and insightful questions.
> >
> > **Q: I/C Comparison**
> >
> > Thanks for the question. The full 100 I/C vanilla BPTT can get 71.8 and the 10 I/C can get 70.1, without extra augmentation and downsampling.
> >
> > **Q: plot of memory budget and bases ratios**
> >
> > Thanks. Note that the bases are also part of the learnable parameters in our model. Increasing the number of bases (learnable) will increase the expressive power in representing images across classes. We agree that it would be very interesting to explore the transformer type of attention and parameterizations. Thank you for the feedback!
> >
> > **Q: momentum term**
> >
> > We did tune the learning rate, and there is a fluctuation in the gradients. We will inspect more. Thank you for the suggestions on forward-only Adam and the second momentum term. They are all great points. We’ll keep exploring this and add them to the paper.
> >
> > **Q: buffer size in Continual Learning**
> >
> > Initially, we were just putting the accuracies of every 4 tasks and the last 4 tasks for better viewing, regarding the question of whether the shrinking buffer sizes will affect the performance on late tasks, but realizing it might be easiest to put all the numbers from all 20 tasks.
> >
> > **Q: related work**
> >
> > Thank you for bringing up the paper. We are aware of this paper and think it could be a great combination to incorporate the TBPTT into our model as the training framework. We will also discuss this work in our paper.

---

### Author Response · Authors · 2022-08-09
**Thanks to all reviewers**

Dear reviewers,

Thank you again for providing feedback and questions on our paper. We will incorporate the discussion so far into our paper, and welcome any additional comments!

In the meantime, we were actually able to update and verify our findings on the higher-resolution TinyImageNet 64x64, which was one of the key lingering concerns. Using the same model backbone (ConvNet with 4 convolutional layers, Instance Norm, and ReLU activations) and a distillation budget of 1 image per class, we can obtain 16.0% (± 0.7) retraining accuracy, significantly outperforming the prior method MTT/TM [15] at 8.8% (± 0.3) accuracy. We’re very excited about this result, and will deepen the exploration and add full results on this benchmark in the next revision.

---

### Meta-Review · Area_Chair_goSM · 2022-08-22

**Recommendation:** Accept
**Confidence:** Certain

**Metareview:**

This paper proposes a new dataset distillation method that achieves SotA results on several benchmarks. Authors were very responsive to answer reviewers' questions, and made significant improvements to the manuscript, also adding additional results confirming the benefits of their approach. At the end of the discussion period, there is a clear consensus for acceptance, due to the fact that this approach is original, well motivated and achieves strong results.

I thus recommend to accept the paper, even though some concerns remain regarding the scalability of the algorithm (in terms of memory usage and running time).

**Award:**

No

---

### Decision · Program_Chairs · 2022-09-14

Accept